# MaSS: Multi-attribute Selective Suppression for Utility-preserving Data Transformation from an Information-theoretic Perspective

## Abstract

The growing richness of large-scale datasets has been a crucial driving force behind the rapid advancement and wide adoption of machine learning technologies. The massive collection and usage of data, however, pose an increasing risk for people's private and sensitive information due to either inadvertent mishandling or malicious exploitation. Besides legislative solutions, many technical approaches have been proposed towards data privacy protection. However, they bear various limitations such as leading to degraded data availability and utility, or relying on heuristics and lacking solid theoretical bases. To overcome these limitations, we propose a formal information-theoretic definition for this utility-preserving privacy protection problem, and design a data-driven learnable data transformation framework that is capable of selectively suppressing sensitive attributes from target datasets while preserving the other useful attributes, regardless of whether or not they are known in advance or explicitly annotated for preservation. We provide rigorous theoretical analyses on the operational bounds for our framework, and carry out comprehensive experimental evaluations using datasets of a variety of modalities, including facial images, voice audio clips, and human activity motion sensor signals. Results demonstrate the effectiveness and generalizability of our method on different tasks and configurations.

## 1 Introduction

The recent rapid advances in machine learning technologies and their wide adoptions through many facets of people's lives are largely attributed to not only the explosive growth in raw computing power, but also the unprecedented availability of large scale datasets, for example, the monumental computer vision dataset ImageNet (Deng et al., 2009), the large multi-lingual web corpus Common Crawl (2023), and the widely used UCI HAR dataset (Anguita et al., 2013). While the vast amount of data serve as the rich basis for machine learning algorithms to learn from, their ubiquitously collection and usage have drawn serious privacy concerns since people's private and sensitive information could be easily leaked through both inadvertent mishandling or deliberate malicious exploitation. Therefore, various regulatory policies, such as GDPR and CCPA, have been drafted and put in place to guardrail the handling and usage of data. While such legislative solutions do generally help mitigate the privacy concerns, they also tend to pose blanket restrictions that result in degraded data availability. Therefore, there has been growing interests in developing more sophisticated and flexible technical/algorithmic solutions.

Towards this goal, many techniques have been proposed to provide data privacy protection. One of the most well-known studies is the protection against membership inference attacks, also known as Differential Privacy (Dwork et al., 2014; Mironov, 2017; Abadi et al., 2016). It focuses on preventing attackers from differentiating between two neighboring sets of samples by observing the change in the distribution of output statistics. Another widely discussed notion of privacy is the protection against attribute inference attacks (Hukkelås et al., 2019; Bertran et al., 2019; Huang et al., 2018; Hsu et al., 2020). This line of work aims at data transformations that remove or suppress sensitive attributes from the dataset while preserving the utility of the dataset for downstream tasks. In our work, we focus our discussion on providing this second type of privacy protection.

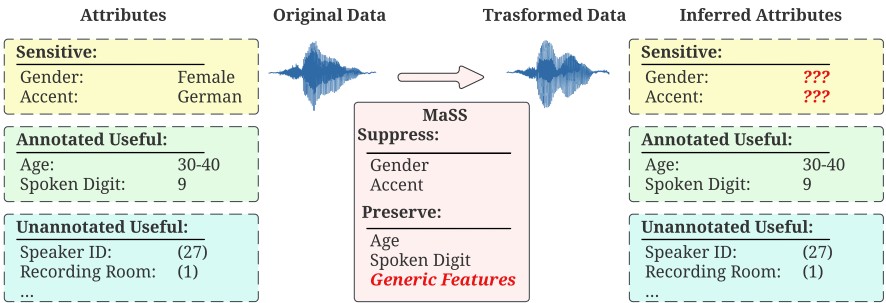

Figure 1: An illustrative use case of MaSS: The original data sample is an voice clip of a person speaking a digit, where its attributes "gender" and "accent" are considered as sensitive, while its "age" and "spoken digit" are annotated as useful. We are also interested in preserving generic features of the data. For example, the voice clip may contain attributes such as "speaker ID" or "recording room" that could prove to be useful down the road, but are not necessarily explicitly annotated yet at the time of processing. After the transformation of MaSS, sensitive attributes can no longer be accurately inferred, but the other useful attributes are preserved in the transformed data.

Various techniques have been studied on balancing between data privacy and utility in defending against attribute inference attacks. However, there have been various limitations in these proposals. For example, Bertran et al. (2019), Wu et al. (2020), and Kumawat & Nagahara (2022) can only ensure the predictability in the transformed data for attributes that have already been explicitly annotated for preservation; no considerations are given to protecting data's unannotated attributes or generic features. On the other hand, Huang et al. (2018), Malekzadeh et al. (2019), and Singh et al. (2022) do account for unannotated attributes, but their designs are mostly heuristic-driven and lack rigorous theoretical bases, which could limit their applicability, especially for scenarios involving highly sensitive information.

To address these limitations, in this paper we present MaSS, a **M**ulti-**a**ttribute **S**elective **S**uppression framework that aims at providing an information-theoretic data transformation solution against attribute inference attacks, with the ability of preserving the utility for not only the annotated but also the unannotated, potentially non-sensitive, yet useful attributes. An illustrative use case of MaSS is shown in Figure 1. We extensively evaluated MaSS on three datasets of different modalities, namely voice recordings, human activity motion sensor signals, and facial images, and show its effectiveness under various configurations of sensitive and useful attribute selections and annotations.

The contributions of this paper are summarized as follows: $i$) We propose MaSS, an information-theoretic data transformation framework capable of selective suppression of multiple sensitive attributes, while preserving other annotated and unannotated useful attributes in the data; $ii$) We provide rigorous theoretical analysis on the design derivation and operational bounds of our proposed multi-attribute data transformation framework; and $iii$) We experimentally evaluate MaSS extensively on voice audio, human activity motion sensor signal, and facial image datasets, and demonstrate its effectiveness and generalizability.

Omitted proofs, details on experiment setups and training, and additional results are included in the Appendix. Code to reproduce our experiments will be made publicly available after review.

## 2 RELATED WORKS

**Privacy-preserving mechanisms.** A privacy-preserving mechanism ensures privacy by randomizing a function of data in order to thwart unwanted inferences. There are two selections of the functions that lead to different privacy notions. If the function is the output of a query over a database, the privacy notion is termed differential privacy (DP) (Dwork et al., 2006), that requires the results of a query be approximately the same for small perturbations of data, and can be usually achieved by additive noise mechanisms (e.g., Gaussian, Laplacian or exponential noise (Dwork et al., 2014; Sun et al., 2020; Zhang et al., 2018; Abadi et al., 2016)). Different from DP, if the function is a conditional distribution that anonymizes sensitive information in the data while preserves non-sensitive information, it leads to the other privacy notion called information-theoretic (IT) privacy. The motivation behind IT privacy is to improve the data quality after anonymization with the additional information of the utility attributes. See Hsu et al. (2021a) for a more detailed discussion on the two privacy notions. Since our goal is to not only suppress the attributes but also preserve the data utility concurrently, the MaSS framework falls within the field of IT privacy.

**IT privacy preserving annotated useful attributes.** Various studies have aimed to anonymize sensitive attributes in data while maintaining its useful information. For instance, Hukkelås et al. (2019) employ a CGAN, conditioned on image background and pose features, to synthesis anonymized facial images. To further ensure de-identification, Maximov et al. (2020) propose to condition the CGAN on an identity control vector, creating images with fabricated identities. Nevertheless, these methods prioritize visual quality of the generated images over the retention of non-sensitive attributes, undermining the data's usefulness for downstream ML tasks. Bertran et al. (2019) counter this by ensuring that explicitly annotated beneficial attributes remain predictable in anonymized data, while thwarting inference of sensitive attributes from an information-theoretic perspective. Wu et al. (2020) extend this approach by introducing a heuristic cross-entropy-based suppression and preservation loss. This idea is further blended with a prior-based suppression loss by Kumawat & Nagahara (2022). Despite these advancements, these studies do not consider the protection of unannotated useful attributes or generic features in the data.

**IT privacy preserving unannotated generic features.** In the endeavor to maintain unannotated attributes, Huang et al. (2018) suggest constraining the distortion of the original data. Malekzadeh et al. (2019) expand on this by combining a heuristic penalty on distortion with information theoretic suppression and preservation losses on annotated attributes. Hsu et al. (2020) advocate for the preservation of generic features by specifically locating and obfuscating information-leaking features. Conversely, Dave et al. (2022) target their work at suppressing the generic features of the data, utilizing contrastive learning technique, while ensuring the predictability of annotated attributes. Despite their practical applicability, these works fall short in providing a robust theoretical foundation regarding the derivation and operational bounds of unannotated attributes protection, casting doubts in scenarios demanding high safety assurances.

**Additional relevant domains.** Other relevant topics also include adversarial fair representation learning for bias reduction in representations (Edwards & Storkey, 2015; Madras et al., 2018) and concept removal techniques for eliminating specific concepts from generative model outputs (Gandikota et al., 2023; Schramowski et al., 2023). However, these techniques were designed with different goals as our work and are not integrated into information theoretic frameworks. Therefore, they are not directly applicable to our task at hand.

## 3 PROBLEM FORMULATION

In this paper, we focus on a multi-attribute dataset comprised of original data $X$, a set of $M$ sensitive attributes $S = (S_1, S_2, \ldots, S_M)$, a set of $N$ annotated useful attributes $U = (U_1, U_2, \ldots, U_N)$, and a set of unannotated useful attributes or generic features $F$. However, our access is limited to the observable joint distribution $P(X, U, S)$, as opposed to the intrinsic joint distribution $P(X, U, S, F)$. We base our work on a pragmatic assumption that $S, U$ are random variables following finite categorical distributions, allowing the mutual information between $S, U$, and $X$ to be bounded. Additionally, we presuppose that with the given $X$, the corresponding annotated attributes $S, U$ are entirely determined (i.e., $P(S_i|X)$ and $P(U_j|X)$ are degenerate distributions). For broad applicability, we do not make assumptions regarding the dimension or distribution family for $F$ and $X$. Furthermore, we do not assume independence between $F$ and other variables, which means that $F$ may correlate with the joint distribution of $X, S, U$.

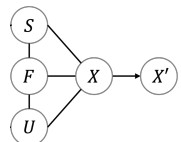

Figure 2: The Markov chain of all variables. $F$ is correlated with $U, S, X$. $X'$ is only dependent on $X$.

Our goal is then to find the optimal data transformation $P_\theta(X'|X)$ and the strongest unannotated useful attribute extractor $P_\eta(F|X')$ by solving the following constrained optimization problem:

$$\max_{\theta, \eta} \quad I(X'; F) \text{ such that } I(X'; S_i) \leq m_i \text{ and } I(X'; U_j) \geq n_j, \tag{1}$$

where $i \in 1 \ldots M$, and $j \in 1 \ldots N$, $P_\theta(X'|X)$ and $P_\eta(F|X')$ are parameterized by neural networks $\theta, \eta$ respectively. By solving this optimization problem, we try to ensure that, at least $n_j$ nats (the counterpart of bits with Napierian base) information is preserved for $U_j$ in the transformed data $X'$, at most $m_i$ nats information is leaked for $S_i$ in $X'$, and the information preserved for $F$ in $X'$ is maximized when the most informative $F$ is extracted from $X$. For clarity, the Markov Chain of variables $U, S, F, X$, and $X'$ corresponding to our problem formulation is illustrated in Figure 2.

### 3.1 OPERATIONAL BOUNDS

To ensure the solvability of the optimization Problem 1, it is essential to demonstrate that the parameters $m_i$ and $n_j$ cannot be chosen arbitrarily.

**Theorem 3.1.** *For the Markov Chain shown in Figure 2, there exists a solution to the optimization problem defined in Equation 1, only if for any pair of $(m_i, n_j)$, $i \in 1 \ldots M$, $j \in 1 \ldots N$, it satisfies:*

$$n_j \leq m_i + I(X; U_j | S_i), \qquad n_j \leq I(X; U_j), \qquad m_i \geq 0. \qquad (2)$$

Under our assumption that $U, S$ are entirely determined given $X$ (i.e., $P(S_i|X)$ and $P(U_j|X)$ are degenerate distributions), Equation 2 can be further simplified to

$$n_j \leq m_i + H(U_j | S_i), \qquad n_j \leq H(U_j), \qquad m_i \geq 0. \qquad (3)$$

Please refer to Appendix A.1 for the proof.

It is important to note that the values in Equation 3, specifically $H(U_j|S_i)$ and $H(U_j)$, are independent of our model's parameters and can be computed prior to training to assess solvability. To elucidate the requirement of $n_j \leq m_i + H(U_j|S_i)$ intuitively, consider a facial image dataset with two attributes "hair color" and "age". The high correlation between these attributes is evident, as older individuals are more likely to have white or gray hair. Should "age" be suppressed with a small $m_{\text{age}}$, the "hair color" information in the facial image must be correspondingly sacrificed to prevent inadvertently disclosing "age" information. The extent of this sacrifice is intuitively determined by the certainty with which "age" predicts "hair color".

Next, we discuss the operational bounds on the optimization objective $I(X'; F)$.

**Theorem 3.2.** *For the Markov Chain shown in Figure 2, for any $m_i$, $i \in 1 \ldots M$, we have*

$$I(X'; F) \leq H(X|S_i) + m_i. \qquad (4)$$

To intuitively understand this, revert to the example we discussed above. When suppressing "age," certain features that was in $X$ no longer reside in $X'$, such as hair color and wrinkles, etc. This results in a necessary sacrifice of the information of $F$ contained in $X'$. The extent of sacrifice is determined by the certainty with which "age" determines the image.

## 4 DATA-DRIVEN LEARNABLE DATA TRANSFORMATION FRAMEWORK

Building upon our problem formulation, we design a learnable data-driven data transformation framework as an approximation to Equation 1, which we call Multi-attribute Selective Suppression (abbreviated as MaSS) as introduced in Section 1. Notably, MaSS is fully differentiable and can be trained using gradient descent to optimize both $P_\theta(X'|X)$ and $P_\eta(F|X)$. The overarching architecture of MaSS is depicted in Figure 3. In the subsequent sections, we elaborate on each of the four modules of MaSS in detail.

### 4.1 DATA TRANSFORMATION

The data transformation module takes in the original data $X$ and outputs the transformed data $X'$. In line with Bertran et al. (2019), we parameterize $P_\theta(X'|X)$ as a neural network $X' = g_\theta(X, a)$, wherein $a$ is a noise variable sampled from a multi-variate unit Gaussian distribution, serving as the source of randomness for $X'$.

### 4.2 SENSITIVE ATTRIBUTES SUPPRESSION

The sensitive attributes suppression module takes as input the transformed data and outputs a suppression loss $L_{s,i}$ for the $i$-th sensitive attribute. The derivation of $L_{s,i}$ is shown as follows. Under the assumption that attributes $S$ can be fully determined given $X$ (i.e., $P(S_i|X)$ is a degenerate distribution), the mutual information $I(X'; S_i)$ can be reformulated as

$$
\begin{aligned}
I(X'; S_i) &= \mathbb{E}_{P(X)P_\theta(X'|X)P(S_i|X)} \left[ \log \frac{P(S_i|X')}{P(S_i)} \right] \\
&= I(X; S_i) - \mathbb{E}_{P(X)P_\theta(X'|X)}[H(P(S_i|X), P(S_i|X'))],
\end{aligned}
\qquad (5)
$$

where $H(\cdot, \cdot)$ denotes cross-entropy, $I(X; S_i)$ can be calculated before training, and the expectation is estimated using mini-batch during training. However, the direct computation of $P(S_i|X')$ is

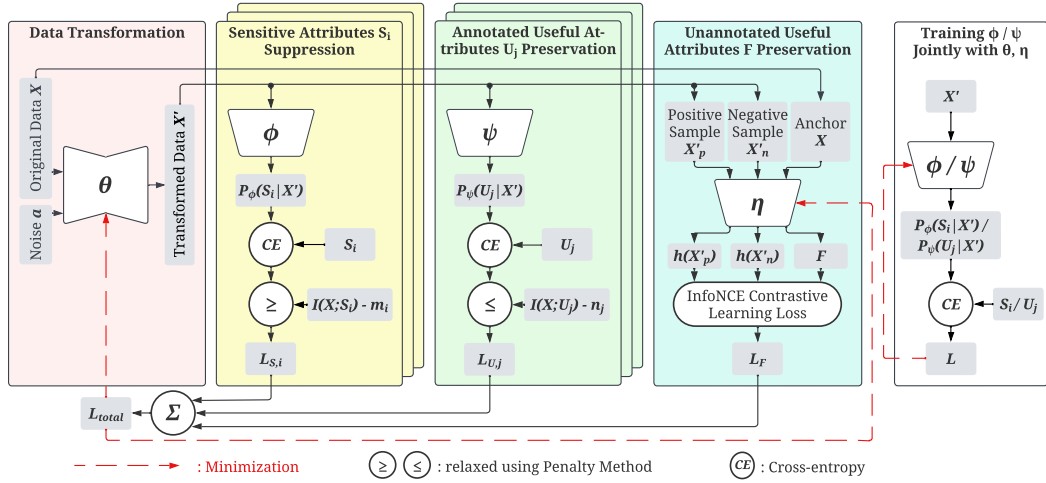

Figure 3: The overall architecture of MaSS. The data transformation module converts the original data into a transformed version. Then the transformed data is sent to both the sensitive attributes suppression module and the annotated useful attributes preservation module, to calculate a relaxed suppression or preservation loss for each attribute respectively. Additionally, the original and transformed data are sent to the unannotated useful attributes preservation module to calculate a contrastive loss. Finally, these losses are aggregated to minimize $\theta$ and $\eta$ jointly. $\phi, \psi$ are optimized with traditional supervised learning.

infeasible because of the intricate nature of neural network $P_\theta(X'|X)$. To address that, we estimate $P(S_i|X')$ with a neural network $P_{\phi_i}(S_i|X')$, which is trained adversarially with $\theta$ using traditional cross-entropy-based supervised learning method:

$$\phi_i = \arg\min_{\phi_i} \mathbb{E}_{P(X)P_\theta(X'|X)}[H(P(S_i|X), P_{\phi_i}(S_i|X'))]. \tag{6}$$

Consequently, the constraint $I(X'; S_i) \leq m_i$ in Equation 1 is converted to

$$I(X; S_i) - m_i \leq \mathbb{E}_{P(X)P_\theta(X'|X)}[H(P(S_i|X), P_{\phi_i}(S_i|X'))]. \tag{7}$$

Following Bertran et al. (2019), we relax the Constraint 7 utilizing the penalty method. Specifically, Equation 7 is converted into a quadratic continuous loss function eligible for gradient descent as:

$$d_{S,i} = \min(\mathbb{E}_{P(X)P_\theta(X|X)}[H(P(S_i|X), P_{\phi_i}(S_i|X'))] + m_i - I(X; S_i), 0)$$
$$L_{S,i} = d_{S,i}^2 + |d_{S,i}| \tag{8}$$

## 4.3 ANNOTATED USEFUL ATTRIBUTES PRESERVATION

The annotated useful attributes preservation module takes as input the transformed data and outputs a preservation loss $L_{u,j}$ for the $j$-th useful attribute. Under the assumption that attributes $U$ can be fully determined given $X$ (i.e., $P(U_j|X)$ is a degenerate distribution), we can reformulate our constraint $I(X'; U_j) \geq n_j$ in a similar way as Section 4.2:

$$I(X'; U_j) = \mathbb{E}_{P(X)P_\theta(X'|X)P(U_j|X)}\left[\log \frac{P(U_j|X')}{P(U_j)}\right] \tag{9}$$
$$= I(X; U_j) - \mathbb{E}_{P(X)P_\theta(X'|X)}[H(P(U_j|X), P(U_j|X'))],$$

where $P(U_j|X')$ is also approximated with neural network $P_{\psi_j}(U_j|X')$. Similar to Section 4.2, $\psi_j$ can be trained jointly (collaboratively) with $\theta$ using traditional supervised learning method:

$$\psi_j = \arg\min_{\psi_j} \mathbb{E}_{P(X)P_\theta(X'|X)}[H(P(U_j|X), P_{\psi_j}(U_j|X'))], \tag{10}$$

and the constraint $I(X'; U_j) \geq n_j$ in Equation 1 can be converted to

$$I(X; U_j) - n_j \geq \mathbb{E}_{P(X)P_\theta(X'|X)}[H(P(U_j|X), P_{\psi_j}(U_j|X'))]. \tag{11}$$

We also convert Equation 11 into the following loss function using quadratic penalty method:

$$d_{U,j} = \max(\mathbb{E}_{P(X)P_\theta(X'|X)}[H(P(U_j|X), P_{\psi_j}(U_j|X'))] + n_j - I(X; U_j), 0)$$
$$L_{U,j} = d_{U,j}^2 + |d_{U,j}|. \tag{12}$$

In order to accelerate the training process, we propose to pre-train an attribute inference network on original data $X$ for each $S_i, U_i$, denoted as $\phi_{i,0}$ and $\psi_{j,0}$ respectively, using traditional supervised learning method. And then we initialize the transformed data attribute inference models $\phi_i$ and $\psi_j$ with $\phi_{i,0}$ and $\psi_{j,0}$ respectively, so that they can converge faster during training.

Note that, different from our method, previous studies such as Bertran et al. (2019) propose to freeze the useful attribute inference model $\psi_j$ during training after it is initialized with $\psi_{j,0}$. Their motivation is that, using this strategy we can potentially learn a space-preserving transformed dataset which shares the same sample space as the original dataset. However, we will show analytically that, this strategy will introduce a noticeable error in estimating the mutual information $I(X'; U_j)$.

**Theorem 4.1.** *Let $P_{\psi_{j,0}}(U_j|X')$ and $I_{\psi_{j,0}}(X'; U_j)$ denote the conditional distribution of $U_j$ given $X'$ and the mutual information between $U_j$ and $X'$ estimated with the **fixed** useful attribute inference network $\psi_{j,0}$. For the Markov Chain shown in Figure 2, we have*

$$I(X'; U_j) - I_{\psi_{j,0}}(X'; U_j) = KL(P(U_j|X')||P_{\psi_{j,0}}(U_j|X')), \tag{13}$$

where $KL(P(U_j|X')||P_{\psi_{j,0}}(U_j|X'))$ can be significantly large and even unbounded, because $\psi_{j,0}$ is trained to approximate $P(U_j|X)$ rather than $P(U_j|X')$. Please refer to Appendix A.3 for proof. Therefore, this strategy is not adopted in our design. However, since learning a space-preserving transformation is still desirable, we propose to achieve this goal in the unannotated useful attributes preservation module.

## 4.4 UNANNOTATED USEFUL ATTRIBUTES PRESERVATION

Without assumptions on the distribution family of $F$ and $X'$, approximating $I(X'; F)$ using methods mentioned above is not feasible. As an alternative, we approximate $I(X'; F)$ using negative InfoNCE contrastive learning loss function as presented in Oord et al. (2018):

$$L_F = \mathbb{E}_{x \sim P(X)}\mathbb{E}_{f \sim P_\eta(F|X)}\mathbb{E}_{x'_p \sim P_\theta(X'|X)}\mathbb{E}_{x'^{(K)}_n \sim P_\theta(X')}[\log \frac{\mathcal{F}(f, x'_p)}{\mathcal{F}(f, x'_p) + \sum_{x'_n \in x'^{(K)}_n} \mathcal{F}(f, x'_n)}] \tag{14}$$

$$I(X'; F) \approx -L_F + \log(K+1), \tag{15}$$

where $f, x'_p, x'_n$ are the anchor, the positive sample, and the negative samples respectively. $K + 1$ is the number of samples including 1 positive sample and $K$ negative samples. $\mathcal{F}$ is a score function defined in the same way as SimCLR (Chen et al., 2020), which can be written as

$$\mathcal{F}(f, x') = e^{\cos(f, h(x'))/\tau}, \tag{16}$$

where $\tau$ is the temperature hyper-parameter. $h(x')$ is a feature extractor trained jointly with $\theta$. Note that unlike SimCLR, our loss do not sample negative samples from $P(F)$.

In order to reduce the number of parameters and hence stabilize the training, we propose to use a single neural network $\eta$ to parameterize both $h_\eta(x')$ and $P_\eta(F|X)$. This symmetric design can also encourage the transformed data $X'$ remains in the same sample space as $X$. Importantly, an alternative interpretation of this design is to apply the InfoNCE contrastive learning loss on $X$ and $X'$ to estimate and maximize $I(X', X)$.

Aligned with pretraining the attribute inference networks, our unannotated useful attributes extractor $\eta$ is also initialized with $\eta_0$ pretrained on the original dataset $X$. In the pretraining stage we use one sample in the mini-batch as both the anchor and the positive sample and use the other samples in the mini-batch as negative samples.

Analogous to Equation 14, we can define another InfoNCE loss of the same form anchored on $X'$:

$$L'_F := \mathbb{E}_{x \sim P(X)}\mathbb{E}_{x' \sim P_\theta(X'|X)}\mathbb{E}_{f_p \sim P_\eta(F|X)}\mathbb{E}_{f^{(K)}_n \sim P_\eta(F)}[\log \frac{\mathcal{F}(f_p, x')}{\mathcal{F}(f_p, x') + \sum_{f_n \in f^{(K)}_n} \mathcal{F}(f_n, x')}] \tag{17}$$

where $x', f_p, f_n$ are the anchor, the positive sample, and the negative samples respectively. Both of these two InfoNCE losses are used for training.

Compared with heuristic reconstruction loss (e.g., MSE), our contrastive learning loss is advantageous in that it is an approximation to $I(X'; F)$, allowing integration into theoretical framework. Additionally, it does not presuppose the distributions of $X, X'$, making it broadly applicable across various domains like images, language, and sensor signals. Moreover, its superior empirical effectiveness is demonstrated in our experiments. A more detailed elaboration of InfoNCE is presented in Section B.1.

## 4.5 MODULE AGGREGATION

Aggregating the losses output from all modules above, we convert our original constrained optimization problem defined in Equation 1 into the following differentiable optimization problem ready for

gradient descent:

$$\min_{\theta,\eta} L_{\text{total}} = \frac{L_F + L'_F}{2} + \lambda \left( \sum_i L_{S,i} + \sum_j L_{U,j} \right) \tag{18}$$

where $\lambda$ is a hyper parameter controlling the degree of relaxation. When $\lambda \to \infty$, Equation 18 recovers the constrained optimization problem defined in Equation 1.

## 5 EVALUATION

In this section, we present our experimental evaluation of MaSS against several baselines methods using multiple datasets of varying modalities.

### 5.1 EXPERIMENTAL SETUP

**Datasets.** The evaluation of MaSS is exhaustively conducted on three multi-attribute benchmark datasets of different modalities, namely the AudioMNIST (Becker et al., 2018) dataset for recorded human voices, the Motion Sense (Malekzadeh et al., 2019) dataset for human activity sensor signals, and the Adience (Eidinger et al., 2014) dataset for facial images. We used the raw data points for training on Motion Sense and Adience, whereas we converted the raw data points to feature embeddings for AudioMNIST using state-of-the-art feature extractor HuBERT-B (Hsu et al., 2021b) for training efficiency.

**Baselines.** We compare our method with 5 baselines, namely ALR (Bertran et al., 2019), GAP (Huang et al., 2018), MSDA (Malekzadeh et al., 2019), BDQ (Kumawat & Nagahara, 2022), and PPDAR (Wu et al., 2020). All 6 methods rely on adversarially training a sensitive attribute inference model. However, ALR, BDQ, and PPDAR do not consider the preservation of unannotated useful attributes, whereas GAP and MSDA do, using a MSE-based heuristic loss. Notwithstanding, GAP does not consider the preservation of annotated useful attribute.

**Evaluation Metrics.** This paper is focused on suppressing sensitive attributes while preserving useful attributes, rather than generating high quality synthetic data. Therefore, we adopt classification accuracy for each attribute on evaluation set as our metric to measure the effectiveness of the suppression or preservation. Specifically, for sensitive attributes, we report the classification accuracy of the adversarially trained classifier $\phi_i$. For useful attributes, to ensure a fair comparison with baselines, we report the classification accuracy of a classifier tuned on the transformed data $X'$ and its attributes $U_j$. The performance is considered better when the sensitive attributes' accuracies are lower and the useful attributes' accuracies are higher.

Furthermore, since the datasets we use are unbalanced, we also report the classification accuracy of the majority classifier as a lower reference value, which can also be interpreted as the accuracy of *guessing* the attribute without accessing $X'$ (Asoodeh et al., 2018; Liao et al., 2019). On the other hand, we also report the accuracy of the $\phi_{i,0}$ and $\psi_{j,0}$ on original data $X$ in the evaluation set as a upper reference value of classification accuracy, which reflects the classification accuracy when no attributes are suppressed.

Based on the lower and upper reference values of classification accuracy, we introduce a noval normalized metric for our task, namely Normalized Accuracy Gain (NAG), which is defined as $\text{NAG} = \max \left( 0, \frac{Acc - Acc_{\text{guessing}}}{Acc_{\text{no\_suppression}} - Acc_{\text{guessing}}} \right)$, where $Acc$ denotes classification accuracy. NAG is

Table 1: Comparison of the classification accuracy and NAG between MaSS and baselines on AudioMNIST. We suppress gender, accent, age, ID, while preserve digit as if an unannotated attribute.

| Method | Accuracy (Normalized Accuracy Gain) | | | | |
|---|---|---|---|---|---|
| | gender ($\downarrow$) | accent ($\downarrow$) | age ($\downarrow$) | ID ($\downarrow$) | digit ($\uparrow$) |
| No suppression | 0.9962 (1.0000) | 0.9843 (1.0000) | 0.9657 (1.0000) | 0.9808 (1.0000) | 0.9975 (1.0000) |
| Guessing | 0.8000 (0.0000) | 0.6833 (0.0000) | 0.1667 (0.0000) | 0.0167 (0.0000) | 0.1000 (0.0000) |
| ALR | 0.8000 (0.0000) | 0.6833 (0.0000) | 0.1667 (0.0000) | 0.0171 (0.0004) | 0.1930 (0.1036) |
| GAP | 0.8000 (0.0000) | 0.6828 (0.0000) | 0.1663 (0.0000) | 0.0438 (0.0281) | 0.9513 (0.9485) |
| MSDA | 0.8000 (0.0000) | 0.6833 (0.0000) | 0.1665 (0.0000) | 0.0238 (0.0074) | 0.9482 (0.9451) |
| BDQ | 0.8000 (0.0000) | 0.6833 (0.0000) | 0.1667 (0.0000) | 0.0275 (0.0112) | 0.5995 (0.5565) |
| PPDAR | 0.8000 (0.0000) | 0.6833 (0.0000) | 0.1667 (0.0000) | 0.0182 (0.0016) | 0.3548 (0.2839) |
| MaSS | 0.8000 (0.0000) | 0.6833 (0.0000) | 0.1667 (0.0000) | 0.0195 (0.0029) | **0.9683 (0.9675)** |

Table 2: Comparison of the classification accuracy and NAG between MaSS and baselines on AudioMNIST. We suppress gender, accent, age, while preserve digit as annotated useful attribute, and preserve ID as if an unannotated attribute.

| Method | Accuracy (Normalized Accuracy Gain) | | | | |
|---|---|---|---|---|---|
| | gender ($\downarrow$) | accent ($\downarrow$) | age ($\downarrow$) | ID ($\uparrow$) | digit ($\uparrow$) |
| No suppression | 0.9962 (1.0000) | 0.9843 (1.0000) | 0.9657 (1.0000) | 0.9808 (1.0000) | 0.9975 (1.0000) |
| Guessing | 0.8000 (0.0000) | 0.6833 (0.0000) | 0.1667 (0.0000) | 0.0167 (0.0000) | 0.1000 (0.0000) |
| ALR | 0.7995 (0.0000) | 0.6832 (0.0000) | 0.1712 (0.0056) | 0.6947 (0.7032) | 0.9970 (0.9994) |
| GAP | 0.8000 (0.0000) | 0.6828 (0.0000) | 0.1663 (0.0000) | 0.6950 (0.7036) | 0.9597 (0.9579) |
| MSDA | 0.8003 (0.0015) | 0.6837 (0.0013) | 0.1925 (0.0323) | 0.8292 (0.8428) | 0.9958 (0.9981) |
| BDQ | 0.8000 (0.0000) | 0.6835 (0.0007) | 0.1677 (0.0013) | 0.4060 (0.4038) | 0.9957 (0.9980) |
| PPDAR | 0.8000 (0.0000) | 0.6833 (0.0000) | 0.1667 (0.0000) | 0.6942 (0.7027) | 0.9960 (0.9983) |
| MaSS | 0.8000 (0.0000) | 0.6833 (0.0000) | 0.1667 (0.0000) | **0.8375 (0.8514)** | 0.9960 (0.9983) |

inherently non-negative, with NAG $= 0$ suggesting that $Acc \leq Acc_{\text{guessing}}$. We consider all $Acc \leq Acc_{\text{guessing}}$ as equally effective, which indicates that this attribute is completely suppressed from $X'$. NAG can be seen as a more informative indicator of how the classification accuracy of each attribute is increased or decreased.

In order to evaluate the performance of MaSS on preserving unannotated useful attributes, we conceal the labels (annotations) of certain annotated attributes during training and only use these labels for evaluation.

**Hyperparameters.** Throughout our experiments, $\lambda$ is simply set to 1. When an attribute is suppressed we simply set its mutual information constraint $m$ as 0. Unless otherwise noted, we set the $n$ of all preserved annotated attributes as the maximal value permitted by Equation 3.

Additional detailed descriptions of the datasets, model structures, and optimization process are elaborated in Appendix C. Next we present and discuss our detailed experimental results.

## 5.2 COMPARISON WITH BASELINES

In this section, we compare MaSS with the 5 baselines described above on the AudioMNIST dataset. The initial experiment involves the suppression of gender, accent, age, and ID attributes while concealing the labels of the digit attribute, treating it as an unannotated attribute for preservation. This setup mirrors scenarios aspiring to eliminate sensitive identity-related information from an audio dataset lacking explicit annotation on non-sensitive attributes. Results of this experiment are shown in Table 1. We can observe that MaSS achieves the highest NAG on digit. In comparison with GAP and MSDA, which also prioritize the preservation of unannotated attributes, MaSS not only attains a higher NAG on digit but also exhibits lower or equal NAG on suppressed attributes. This outcome underscores the limitation of the MSE-based heuristic loss used in GAP and MSDA for unannotated useful attributes preservation, as it overly restricts the flexibility of data transformation.

In the subsequent experiment, we aim to suppress gender, accent, and age, while preserving digit as annotated and ID as unannotated. This scenario emulates conditions wherein the dataset encompasses both sensitive and useful annotated attributes, alongside with to-be-preserved unannotated attributes. The results are shown in Table 2. It is observable that MaSS secures the highest NAG on ID, along with a NAG on digit that is comparably high to other methods. Notably, even though MSDA's NAG on ID is close to MaSS, it adversely bears a higher NAG across all suppressed attributes. Moreover, GAP falls short in achieving a high NAG on digit relative to other methods as it does not specifically account for the preservation of annotated useful attributes.

We further compared our method with SPAct, the results of which is shown in Appendix D.1. These experimental results demonstrate that our method is robust and superior in the overall performance for simultaneous privacy suppression and utility preservation for both annotated and unannotated attributes.

Next, the application of MaSS is extended to the human activity sensor signal dataset, Motion Sense. Initial experiments focus on suppressing gender and ID attributes, while treating activity as an unannotated attribute for preservation. Results as shown in Table 3 demonstrate that MaSS attains the highest NAG on the activity attribute. Additionally, in comparison to GAP and MSDA, our method showcases a superior NAG on activity and a reduced NAG on both suppressed attributes. This results further substantiates MaSS's proficiency in maintaining a superior balance between preserving meaningful features and effectively suppressing sensitive attributes.

Table 3: Comparison of the classification accuracy and NAG between MaSS and baselines on Motion Sense. We suppress gender, ID, while preserve activity as if unannotated useful attribute.

| Method | Accuracy (Normalized Accuracy Gain) | | |
|---|---|---|---|
| | gender (↓) | ID (↓) | activity (↑) |
| No suppr. | 0.9817 (1.0000) | 0.9026 (1.0000) | 0.9764 (1.0000) |
| Guessing | 0.5699 (0.0000) | 0.0533 (0.0000) | 0.4663 (0.0000) |
| ALR | 0.6040 (0.0828) | 0.0900 (0.0432) | 0.8593 (0.7704) |
| GAP | 0.5721 (0.0053) | 0.0800 (0.0314) | 0.8937 (0.8379) |
| MSDA | 0.5725 (0.0063) | 0.1134 (0.0708) | 0.8957 (0.8418) |
| BDQ | 0.6184 (0.1178) | 0.1054 (0.0613) | 0.8451 (0.7426) |
| PPDAR | 0.5698 (0.0000) | 0.0498 (0.0000) | 0.8189 (0.6912) |
| MaSS | 0.5686 (0.0000) | 0.0555 (0.0026) | **0.9242 (0.8977)** |

Table 4: Comparison of the classification accuracy and NAG between MaSS and baselines on Adience. We suppress gender, while preserve age, ID as if unannotated useful attributes.

| Method | Accuracy (Normalized Accuracy Gain) | | |
|---|---|---|---|
| | gender (↓) | age (↑) | ID (↑) |
| No suppr. | 0.9774 (1.0000) | 0.9321 (1.0000) | 0.9382 (1.0000) |
| Guessing | 0.5240 (0.0000) | 0.2892 (0.0000) | 0.0284 (0.0000) |
| ALR | 0.5298 (0.0128) | 0.2907 (0.0023) | 0.0400 (0.0128) |
| GAP | 0.5240 (0.0000) | 0.6047 (0.4907) | 0.5393 (0.5616) |
| MSDA | 0.6652 (0.3114) | 0.7989 (0.7928) | 0.7982 (0.8461) |
| BDQ | 0.5252 (0.0026) | 0.2892 (0.0000) | 0.0352 (0.0075) |
| PPDAR | 0.5231 (0.0000) | 0.2892 (0.0000) | 0.0284 (0.0000) |
| MaSS | **0.5240 (0.0000)** | 0.7661 (0.7418) | 0.7255 (0.7662) |

Table 5: Comparison of the NAG for different configurations of MaSS on AudioMNIST. ✓ denotes that this attribute is suppressed, while all other attributes are preserved as annotated useful attributes.

| Method | MaSS Suppressed Attributes | | | | | Normalized Accuracy Gain | | | | |
|---|---|---|---|---|---|---|---|---|---|---|
| | gender | accent | age | ID | digit | gender | accent | age | ID | digit |
| No suppression | | | | | | 1.0000 | 1.0000 | 1.0000 | 1.0000 | 1.0000 |
| Guessing | ✓ | ✓ | ✓ | ✓ | ✓ | 0.0000 | 0.0000 | 0.0000 | 0.0000 | 0.0000 |
| MaSS | ✓ | | | | | 0.0000 | 0.9342 | 0.9574 | 0.9632 | 0.9972 |
| | ✓ | ✓ | | | | 0.0000 | 0.0000 | 0.9199 | 0.9372 | 0.9987 |
| | ✓ | ✓ | ✓ | | | 0.0000 | 0.0000 | 0.0000 | 0.8680 | 0.9964 |
| | ✓ | ✓ | ✓ | ✓ | | 0.0000 | 0.0000 | 0.0000 | 0.0017 | 0.9953 |

We further experiment with suppressing gender, while preserving ID as annotated, and preserving activity as unannotated. Please refer to Appendix D.2 for results and corresponding analysis.

Finally, we apply MaSS to Adience, suppressing gender while treating age and activity as unannotated attributes that should be preserved. The results shown in Table 4 reveals that, among all methods with NAG = 0 for gender, MaSS accomplishes the highest NAG for the preserved attributes. Visualized transformed images, together with additional results on suppressing age are shown in Appendix D.3.

## 5.3 ABLATION STUDY

In ablation study, we first experiment with different configuration of suppressed and preserved attributes using MaSS on AudioMNIST. The configurations and their corresponding results are shown in Table 5. We can see that MaSS consistently achieves NAG = 0 for most of the suppressed attributes, alongside with high NAG for preserved attributes.

We also examined MaSS by replacing contrastive learning loss to MSE reconstruction loss, and with varying $m$ for sensitive attributes. Besides, we also empirically show that the transformed facial images can be accurately used by pre-trained landmark detection model PIPNet Jin et al. (2021). Please refer to Appendix D.4 and D.5 for detailed results.

## 6 CONCLUSION

In this paper, we present MaSS, a generalizable and highly configurable data-driven learnable data transformation framework that is capable of suppressing sensitive/private information from data while preserving its utility. Compared to existing privacy protection techniques that have similar objectives, MaSS overcomes their limitations by possessing the following two characteristics: $i$) It provides a contrastive-learning-based mechanism that preserves not only explicitly annotated, known-in-advance useful attributes of the data but also potentially all other useful attributes that have not necessarily been considered or annotated at the time of processing, thus greatly maximizing data utility and applicability for downstream analytical tasks; and $ii$) As opposed to being purely heuristic-driven, MaSS is built on top of rigorous information-theoretic bases with clear objective and operational bounds. We thoroughly evaluated MaSS on three datasets of different modalities, namely voice recordings, human activity motion sensor signals, and facial images, and obtained promising results that demonstrate MaSS' effectiveness under various tasks and configurations.

**Ethics statement.**   We believe that there is no ethical concern related to this work. Our work benefits the protection of people's privacy in that it is proposed to suppress sensitive attributes in the datasets while preserve their potential utility for downstream tasks.

**Reproducibility statement.**   For experiments, we provide detailed descriptions of the datasets, pre-processing steps, model structures, optimization procedures, hyperparameter settings used in our experiments in Section 5.1 and Appendix C. For theoretical works, we include detailed proofs in Appendix A.

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

APPENDIX

# A PROOFS

## A.1 PROOF OF THEOREM 3.1

*Proof.* For the Markov Chain shown in Figure 2, for any $i \in 1 \ldots M$, and $j \in 1 \ldots N$, if both $I(X'; S_i) \leq m_i$ and $I(X'; U_j) \geq n_j$ hold, then we have

$$
\begin{aligned}
m_i + I(X; U_j | S_i) &\geq I(X'; S_i) + I(X; U_j | S_i) \\
&= I(X'; S_i) + I(X', X; U_j | S_i) \\
&= I(X'; U_j, S_i) - I(X'; U_j | S_i) + I(X', X; U_j | S_i) \\
&= I(X'; U_j, S_i) + I(X; U_j | X', S_i) \\
&= I(X'; U_j) + I(X'; S_i | U_j) + I(X; U_j | X', S_i) \\
&\geq I(X'; U_j) \\
&\geq n_j
\end{aligned}
\tag{19}
$$

which proves the first inequation. Following Data Processing Inequality, we can also have

$$
n_j \leq I(X'; U_j) \leq I(X; U_j)
\tag{20}
$$

which proves the second inequation. Finally, we also have

$$
m_i \geq I(X'; S_i) \geq 0
\tag{21}
$$

which proves the third inequation.

Under the assumption that $U, S$ are fully determined given $X$ ($P(S_i|X), P(U_j|X)$ are one-hot vectors), we can have

$$
H(S_i|X) = 0, \qquad H(U_j|X) = 0
\tag{22}
$$

for any $i \in 1 \ldots M$, and $j \in 1 \ldots N$. Since adding a condition can not increase the entropy, we can also have

$$
0 \leq H(U_j|X, S_i) \leq H(U_j|X) = 0
\tag{23}
$$

Therefore we have

$$
H(U_j|X, S_i) = 0
\tag{24}
$$

Inserting $H(U_j|X, S_i) = 0$ and $H(U_j|X) = 0$ into the inequations 2, we can further convert them to the inequations 3 as

$$
\begin{aligned}
n_j &\leq m_i + I(X; U_j | S_i) \\
&= m_i + H(U_j | S_i) - H(U_j | X, S_i) \\
&= m_i + H(U_j | S_i),
\end{aligned}
\tag{25}
$$

and

$$
\begin{aligned}
n_j &\leq I(X; U_j) \\
&= H(U_j) - H(U_j | X) \\
&= H(U_j).
\end{aligned}
\tag{26}
$$

$\square$

## A.2 PROOF OF THEOREM 3.2

*Proof.* For the Markov Chain shown in Figure 2, according to Data Processing Inequality, we have

$$
I(X'; X) - I(X'; F) = I(X'; X | F) \geq 0
\tag{27}
$$

Therefore, we have

$$
I(X'; F) \leq I(X'; X)
\tag{28}
$$

We can also have

$$
\begin{aligned}
I(X'; X) &= H(X') - H(X'|X) \\
&= H(X') - H(X'|X, S_i) \\
&\leq H(X') - H(X'|X, S_i) + H(X|X', S_i) \\
&= H(X') + H(X|S_i) - H(X'|S_i) \\
&= I(X'; S_i) + H(X|S_i) \\
&\leq H(X|S_i) + m_i
\end{aligned}
\tag{29}
$$

$\square$

A.3   PROOF OF THEOREM 4.1

*Proof.* For the Markov Chain shown in Figure 2, the mutual information between $U_j$ and $X'$ estimated using the **fixed** useful attribute inference network $\psi_{j,0}$, namely $I_{\psi_{j,0}}(X'; U_j)$, is calculated by replacing the $P(U_j|X')$ in Equation 9 with the estimated conditional distribution $P_{\psi_{j,0}}(U_j|X')$ as

$$I_{\psi_{j,0}}(X'; U_j) = \mathbb{E}_{P(X)P_\theta(X'|X)P(U_j|X)}[\log \frac{P_{\psi_{j,0}}(U_j|X')}{P(U_j)}] \tag{30}$$

Therefore, we can have

$$I(X'; U_j) - I_{\psi_{j,0}}(X'; U_j) = \mathbb{E}_{P(X)P_\theta(X'|X)P(U_j|X)}[\log \frac{P(U_j|X')}{P_{\psi_{j,0}}(U_j|X')}]$$

$$= \mathbb{E}_{P_\theta(X')P(U_j|X')}[\log \frac{P(U_j|X')}{P_{\psi_{j,0}}(U_j|X')}] \tag{31}$$

$$= KL(P(U_j|X')||P_{\psi_{j,0}}(U_j|X'))$$

$\square$

# B   ADDITIONAL DESCRIPTION OF PROPOSED METHOD

## B.1   INFONCE CONTRASTIVE LEARNING

InfoNCE contrastive learning loss Oord et al. (2018) is a classical contrastive learning loss, which learns useful representations of data by making the representations of positive samples (similar or related samples) closer while pushing the representations of negative samples further apart from the anchor. The sampling strategy in our framework is as follows. Suppose we have $K + 1$ samples $\{x_i\}_{i=1}^{K+1}$ in a mini-batch. We first pass them through the feature extractor $P_\eta(F|X)$ and data transformation module to sample a batch of $\{f_i\}_{i=1}^{K+1}$ and $\{x'_i\}_{i=1}^{K+1}$ respectively. Then suppose we choose the $j$-th feature $f_j$ as the anchor. Then the corresponding $x'_j$ would be designated as positive sample, and all other $x'_{i\neq j}$ are designated as negative samples. After sampling, we calculate the contrastive learning loss as Equation 14 in our paper. For training stability, in our implementation each of $K + 1$ features in a batch are used as anchor once and then averaged. An analogous strategy was used when anchors are chosen from $x'$.

# C   ADDITIONAL EXPERIMENTAL SETUP

## C.1   DATASETS

We next introduce the datasets. The Adience dataset, consisting of 26580 facial images, was originally published to help study the recognition of age and gender. Each face image has 3 attributes: ID, gender and age. We filter out the IDs with only one image. For the rest of data points, we split them into training and evaluation set as 7:3, and ensure that for each ID there is at least one image in training set and one image in evaluation set. Data points used in our experiment contains 1042 different DataIDs, 8 age groups, and 2 gender classes. The images are resized to 80*80, converted to grayscale images, and normalzied to 0-1 in our experiments.

The AudioMNIST dataset contains audio recordings of spoken digits (0-9) in English from 60 speakers. The dataset contains 8 attributes, from which we used 5 most representative attributes for our experiments, namely gender, accent, age, ID, spoken digits, with 2, 16, 18, 60, 10 classes, respectively. There are 30,000 audio clips in total. We split the data into 24000, 6,000 for training and evaluation. The audio data are transformed to feature embeddings by HuBERT-B feature extractor and normalized to unit L2-norm.

The Motion Sense dataset contains the accelerometer and gyroscope data for human doing 6 daily activities. It contains 5 attributes, form which we used 3 most representative attributes for our experiments, namely gender, ID, and activity, with 2, 24, 6 classes respectively. Following Malekzadeh et al. (2019), we did not use "sit" and "stand up" activity in experiments. We used the same split and data pre-processing method as Malekzadeh et al. (2019), which resulted in 74324 segmented data points. Specifically, we used "trail" split strategy as described in Malekzadeh et al. (2019), and we

Table 6: Model structures and optimization methods used for our experiments.

| Experiment | Audio | Human activity | Facial image |
|---|---|---|---|
| Dataset | AudioMNIST | Motion Sense | Adience |
| # total data points | 30000 | 74324 | 26580 |
| Training-evaluation split | 4:1 | 7:4 | 7:3 |
| Optimizer | AdamW (Loshchilov & Hutter, 2017) | | |
| Learning rate | 0.0001 | | |
| Weight decay | 0.001 | | |
| Learning rate scheduler | Cosine scheduler | | |
| Epoch | 2000 | 200 | 4000 |
| $\theta$ model structure | 3-layer MLP | 3-layer MLP | U-Net |
| $\phi, \psi, \eta$ model structure | 3-layer MLP | 6-layer Convolutional NN | Fixed FaceNet backbone followed by learnable 3-layer MLP |

Table 7: Comparison of the classification accuracy and NAG between MaSS and SPAct on AudioMNIST. We suppress gender, accent, age, id, while preserve digit as annotated useful attribute.

| Method | Accuracy (Normalized Accuracy Gain) | | | | |
|---|---|---|---|---|---|
| | gender ($\downarrow$) | accent ($\downarrow$) | age ($\downarrow$) | ID ($\downarrow$) | digit ($\uparrow$) |
| No suppression | 0.9962 (1.0000) | 0.9843 (1.0000) | 0.9657 (1.0000) | 0.9808 (1.0000) | 0.9975 (1.0000) |
| Guessing | 0.8000 (0.0000) | 0.6833 (0.0000) | 0.1667 (0.0000) | 0.0167 (0.0000) | 0.1000 (0.0000) |
| SPAct | 0.8087 (0.0442) | 0.6833 (0.0001) | 0.1753 (0.0108) | 0.0707 (0.0560) | 0.9948 (0.9970) |
| MaSS | 0.8000 (0.0000) | 0.6833 (0.0000) | 0.1662 (0.0000) | 0.0183 (0.0017) | 0.9933 (0.9953) |

only used the magnitude of gyroscope and accelerometer as input. Signals are normalized to 0-mean and 1-std, and then cut into 128-length clips.

## C.2 MODEL STRUCTURES AND OPTIMIZATION

We elaborated the model structures and optimization methods used for our experiments in Table 6. For faster convergence and training stability, we design the $\phi, \psi, \eta$ models used in facial image experiments as a fixed FaceNet (Schroff et al., 2015) backbone followed by learnable 3-layer MLPs, and design the $\theta$ model of facial image experiment as U-Net (Ronneberger et al., 2015). For the same reason, we add residual structures from input of the first layer to the output of the second layer for 3-layer MLP $\theta$ models used in audio and human activity experiments.

# D ADDITIONAL EXPERIMENTS RESULTS

## D.1 EVALUATION ON HUMAN ACTIVITY SENSOR SIGNALS

We also compare our method with SPAct Dave et al. (2022). Since SPAct did not consider preserving unannotated useful attributes. Therefore we compare it in a scenerio where we only have annotated attributes. We can observe that MaSS achieved slightly lower NAG on digit compared with SPAct, but significantly lower NAG on all sensitive attributes (up to 5%), which shows that our method may achieve a better trade-off between suppression and preservation.

## D.2 EVALUATION ON HUMAN ACTIVITY SENSOR SIGNALS

In this experiment, we suppress gender, while preserve ID as annotated attribute, and preserve activity as unannotated attribute. We set the $n$ for ID as 1.6, which meets the requirements of Equation 3. The results are shown in Table 8. We can observe that MaSS achieved lowest NAG on gender as well as comparable NAG on the other preserved attributes. This outcome stems from the fact the

Table 8: Comparison of the classification accuracy and NAG between MaSS and baselines on Motion Sense. We suppress gender, while preserve ID as annotated useful attribute, and preserve activity as if an unannotated attribute.

| Method | Accuracy (Normalized Accuracy Gain) | | |
|---|---|---|---|
| | gender ($\downarrow$) | ID ($\uparrow$) | activity ($\uparrow$) |
| No suppression | 0.9817 (1.0000) | 0.9026 (1.0000) | 0.9764 (1.0000) |
| Guessing | 0.5699 (0.0000) | 0.0533 (0.0000) | 0.4663 (0.0000) |
| MaSS | **0.5870 (0.0415)** | 0.5931 (0.6356) | 0.9168 (0.8832) |
| ALR | 0.8258 (0.6214) | 0.6147 (0.6610) | 0.8966 (0.8436) |
| GAP | 0.6599 (0.2186) | 0.6628 (0.7176) | 0.9378 (0.9243) |
| MSDA | 0.6418 (0.1746) | 0.6360 (0.6861) | 0.9030 (0.8561) |
| BDQ | 0.7092 (0.3383) | 0.6583 (0.7124) | 0.9269 (0.9030) |
| PPDAR | 0.7830 (0.5175) | 0.5680 (0.6060) | 0.8867 (0.8242) |

Table 9: Comparison of the Accuracy and NAG for different configurations of MaSS on Adience. ✓ denotes that this attribute is suppressed, while all other attributes are preserved as unannotated useful attributes.

| Method | MaSS Suppressed Attributes | | | Accuracy (Normalized Accuracy Gain) | | |
|---|---|---|---|---|---|---|
| | gender | age | ID | gender | age | ID |
| No suppression | | | | 0.9774 (1.0000) | 0.9321 (1.0000) | 0.9382 (1.0000) |
| Guessing | ✓ | ✓ | ✓ | 0.5240 (0.0000) | 0.2892 (0.0000) | 0.0284 (0.0000) |
| MaSS | ✓ | | | 0.5240 (0.0000) | 0.7661 (0.7418) | 0.7255 (0.7662) |
| | | ✓ | | 0.7985 (0.6054) | 0.2892 (0.0000) | 0.5005 (0.5189) |

sensitive attribute gender is determined by ID, therefore when we suppress gender, the information retained for ID is inherently limited as Equation 3. MaSS is explicitly aware of this limit and is adjusted to preserve only limited amount of information for ID. In contrast other baselines can only heuristically trade-off between suppressing and preserving.

### D.3   EVALUATION ON FACIAL IMAGES

In this experiment we demonstrate the performance of MaSS on Adience with different attribute to suppress. We can observe from Table 9 that MaSS achieved 0 NAG for suppressed attributes as well as acceptable NAG for preserved unannotated attributes.

The visualization results for both original and transformed data in the Adience dataset are depicted in Figure 4. Observing the second row, we can see that the gender information has been effectively removed from the images. Similarly, the third row demonstrates the removal of age information from the images, highlighting the efficacy of our approach in suppressing specific attributes.

### D.4   ABLATION STUDY

We conducted ablation experiments on unannotated attributes preservation module, where we remove the unannotated useful attributes preservation module (denoted as MaSS-NF), or replace the contrastive learning loss with MSE reconstruction loss (denoted as MaSS-MSE). The results are shown in Table 10 and Table 11. We can observe that the NAG (normalized accuracy gain) of digit is significantly lower without unannotated attribute preservation. We can also observe that MaSS achieved higher NAG on digit than MaSS-MSE, as well as lower NAG on other sensitive attributes, which demonstrated the strong empirical performance of contrastive learning loss compared to MSE reconstruction loss.

We also conducted experiments to show the effect of varying the constraint on sensitive attributes suppression ($m$). We take gender, accent, age and ID as sensitive attributes and take digit as annotated useful attribute on the AudioMNIST dataset. We fix $m = 0$ for gender, accent and age and

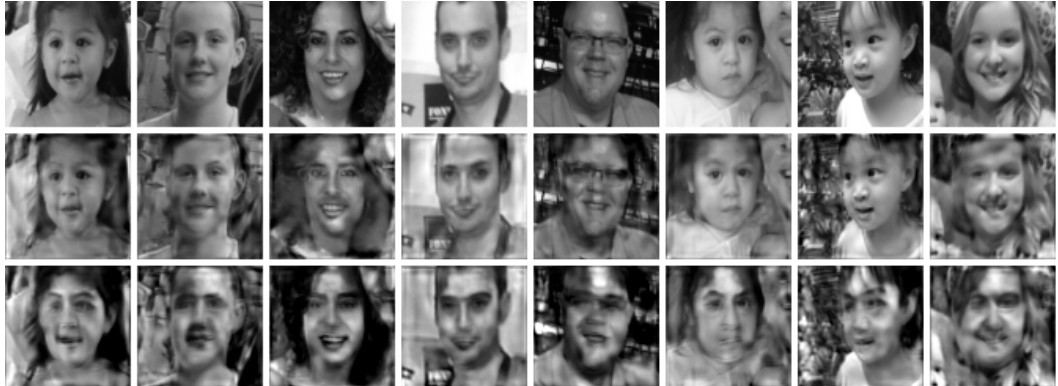

Figure 4: The visualization of the original data and transformed data in Adience dataset. The first row presents the original facial images, while the second and third rows show the transformed image with gender and age suppressed respectively. Other attributes are preserved as unannotated.

Table 10: Comparison of the classification accuracy and NAG between MaSS and ablations on AudioMNIST. We suppress gender, accent, age, ID, while preserve digit as if an unannotated attribute.

| Method | Accuracy (Normalized Accuracy Gain) | | | | |
|---|---|---|---|---|---|
| | gender ($\downarrow$) | accent ($\downarrow$) | age ($\downarrow$) | ID ($\downarrow$) | digit ($\uparrow$) |
| No suppr. | 0.9962 (1.0000) | 0.9843 (1.0000) | 0.9657 (1.0000) | 0.9808 (1.0000) | 0.9975 (1.0000) |
| Guessing | 0.8000 (0.0000) | 0.6833 (0.0000) | 0.1667 (0.0000) | 0.0167 (0.0000) | 0.1000 (0.0000) |
| MaSS-NF | 0.8000 (0.0000) | 0.6833 (0.0001) | 0.1658 (0.0000) | 0.0152 (0.0000) | 0.2657 (0.1846) |
| MaSS-MSE | 0.8002 (0.0008) | 0.6833 (0.0001) | 0.1683 (0.0020) | 0.0462 (0.0306) | 0.9542 (0.9517) |
| MaSS | 0.8000 (0.0000) | 0.6833 (0.0000) | 0.1667 (0.0000) | 0.0195 (0.0029) | **0.9683 (0.9675)** |

Table 11: Comparison of the classification accuracy and NAG between MaSS and ablations on MotionSense. We suppress gender, ID, while preserve activity as if unannotated useful attribute.

| Method | Accuracy (Normalized Accuracy Gain) | | |
|---|---|---|---|
| | gender ($\downarrow$) | ID ($\downarrow$) | activity ($\uparrow$) |
| No suppr. | 0.9026 (1.0000) | 0.9817 (1.0000) | 0.9764 (1.0000) |
| Guessing | 0.0533 (0.0000) | 0.5699 (0.0000) | 0.4663 (0.0000) |
| MaSS-NF | 0.0508 (0.0000) | 0.5699 (0.0000) | 0.8374 (0.7275) |
| MaSS-MSE | 0.0754 (0.0260) | 0.5734 (0.0085) | 0.8823 (0.8156) |
| MaSS | 0.0555 (0.0026) | 0.5686 (0.0000) | **0.9242 (0.8977)** |

$n = 2.3$ for digit (its maximal value). Then we vary $m$ for ID from 0 to 1.46 (its maximal value). The results are shown in Table 12. We can observe that as $m$ increases, the constraint is gradually loosened, which results in the gradually increasing accuracy and NAG for ID.

Although we would not release the labels of sensitive attributes to the public, here we conducted an ablation experiment with the assumption that the attacker can access the ground truth labels of sensitive attributes as an oracle and retrain the discriminator on transformed data. The results are shown in Table 2. We can observe that, using MaSS, the accuracy of the retrained discriminator is higher than adversarial discriminator but is still significantly lower than the discriminator trained using original data.

## D.5 USABILITY FOR PRE-TRAINED MODELS

One of the advantage of releasing transformed data instead of compact representations, is that transformed data can be readily used by pre-trained off-the-shelf models, whereas compact representations can only be used by models specially trained for them. To empirically support this claim,

Table 12: Varying the suppression constraint $m$ for ID on AudioMNIST. We suppress gender, accent, age, ID, while preserve digit as if an annotated useful attribute.

| Method | $m_{ID}$ | Accuracy (Normalized Accuracy Gain) | | | | |
|---|---|---|---|---|---|---|
| | | gender ($\downarrow$) | accent ($\downarrow$) | age ($\downarrow$) | ID ($\downarrow$) | digit ($\uparrow$) |
| No suppression | - | 0.9962 (1.0000) | 0.9843 (1.0000) | 0.9657 (1.0000) | 0.9808 (1.0000) | 0.9975 (1.0000) |
| Guessing | - | 0.8000 (0.0000) | 0.6833 (0.0000) | 0.1667 (0.0000) | 0.0167 (0.0000) | 0.1000 (0.0000) |
| | 0.0 | 0.8000 (0.0000) | 0.6833 (0.0000) | 0.1662 (0.0000) | 0.0183 (0.0017) | 0.9933 (0.9953) |
| | 0.3 | 0.8000 (0.0000) | 0.6833 (0.0000) | 0.1665 (0.0000) | 0.0598 (0.0447) | 0.9938 (0.9959) |
| MaSS | 0.6 | 0.8000 (0.0000) | 0.6833 (0.0000) | 0.1668 (0.0002) | 0.1120 (0.0988) | 0.9940 (0.9961) |
| | 0.9 | 0.8000 (0.0000) | 0.6833 (0.0000) | 0.1670 (0.0004) | 0.1493 (0.1376) | 0.9937 (0.9957) |
| | 1.2 | 0.8000 (0.0000) | 0.6833 (0.0000) | 0.1667 (0.0000) | 0.1963 (0.1863) | 0.9928 (0.9948) |
| | 1.46 | 0.8000 (0.0000) | 0.6833 (0.0000) | 0.1667 (0.0000) | 0.2597 (0.2520) | 0.9937 (0.9957) |

Table 13: Comparison of the accuracy and NAG between a trained-from-scratch discriminator and adversarial discriminator on the Adience dataset. We suppress gender, while preserve age, ID as if unannotated useful attributes.

| Method | Accuracy (Normalized Accuracy Gain) |
|---|---|
| | gender ($\downarrow$) |
| No suppr. | 0.9774 (1.0000) |
| Guessing | 0.5240 (0.0000) |
| MaSS (discriminator retrained with oracle) | 0.6029 (0.1740) |
| MaSS (adversarial discriminator) | 0.5240 (0.0000) |

Table 14: The NME(%) of PIPNet between transformed Adience and original Adience, in comparison with the NME(%) of PIPNet between original WLFW dataset and ground truth label. The comparable performance showed that transformed Adience dataset can be accurately exploited by pre-trained PIPNet. On the contrary, compact representations would not be compatible with pre-trained PIPNet.

| Dataset | WLFW (original vs. label) | Adience (transformed vs. original) |
|---|---|---|
| NME ($\downarrow$) | 3.94 | 3.30 |

we ran an experiment on the landmark detection on both original and transformed Adience datasets based on torchlm (https://github.com/DefTruth/torchlm), with the PIPNet (Jin et al., 2021) with ResNet18 backbone. This model generates 98 landmarks for each facial image. We then take the landmarks detected from the original images as the ground truth and evaluate the landmarks detected from the transformed images, using normalized mean error (NME) (Jin et al., 2021) as quantitative metric. The results are shown in Table 14. We can observe that NME between transformed Adience and original Adience is comparable with the NME between original WLFW dataset (Wu et al., 2018) and ground truth label, which suggests that the transformed dataset can be accurately exploited by a pre-trained model.

