# OpenReview forum: "MaSS: Multi-attribute Selective Suppression for Utility-preserving Data Transformation from an Information-theoretic Perspective"
_ICLR.cc/2024/Conference — Submitted to ICLR 2024_

### Official Review · Reviewer_CAg3 · 2023-10-30

**Soundness:** 3 good
**Presentation:** 4 excellent
**Contribution:** 3 good
**Rating:** 6
**Confidence:** 4

**Summary:**

This paper proposes a framework called MASS (Multi-Attribute Selective Suppression) for privacy-preserving data transformation that selectively suppresses sensitive attributes while preserving useful ones. The authors provide a formal definition of privacy protection and an information-theoretic perspective on the problem. They also present a data-driven approach that uses a combination of supervised and unsupervised learning to identify sensitive attributes and suppress them while preserving useful ones. The authors provide rigorous theoretical analyses and comprehensive experimental evaluations that demonstrate the effectiveness of their approach. The contributions of this paper include a formal definition of privacy protection, a data-driven framework for privacy-preserving data transformation, and a comprehensive evaluation of the proposed approach against several baseline methods using multiple datasets of varying modalities.

**Strengths:**

S1. In terms of originality, the paper introduces a novel approach for protecting unannotated attributes in datasets. While previous works have focused on protecting annotated attributes or using heuristics, the paper proposes a data-driven learnable data transformation framework called MaSS (Multi-Attribute Selective Suppression) that can selectively suppress sensitive attributes while preserving other useful attributes, regardless of whether they are known in advance or explicitly annotated. This approach is unique and addresses a gap in the existing literature.
S2. The quality of the paper is high, as it provides rigorous theoretical analyses of the operational bounds of the proposed framework. The authors derive mathematical formulations and provide proofs for the theorems presented in the paper [5, 7]. This demonstrates a strong understanding of the underlying principles and ensures the reliability of the proposed methods.
S3. The clarity of the paper is commendable. The authors provide clear explanations of the problem formulation, the proposed techniques, and the evaluation methodology. The paper is well-structured, making it easy for readers to follow the flow of ideas. Additionally, the authors provide visualizations and tables to support their findings.

**Weaknesses:**

W1. The dataset lacks a detailed description. It would also be good to have a table that describes the size of the dataset along with some other information that would give the reader a clearer picture of the dataset.
W2. All six methods of experimental comparison rely on adversarially training a sensitive attribute inference model and lack the ability to compare state-of-the-art dp-based methods (e.g. "Mingxuan Sun, Qing Wang, Zicheng Liu: Human Action Image Generation with Differential Privacy. ICME 2020: 1-6").
W3. Lack of comparison with a state-of-the-art method ("Li M, Xu X, Fan H, et al. STPrivacy: Spatio-Temporal Privacy-Preserving Action Recognition[C]//Proceedings of the IEEE/CVF International Conference on Computer Vision. 2023: 5106-5115.")

**Questions:**

The authors need provide more detailed explanations and justifications for their proposed techniques. why contrastive learning is suitable for protecting unannotated attributes and how it ensures the predictability of annotated attributes? Additionally, the authors could consider providing more detailed explanations of the loss functions used in their method, such as the InfoNCE Contrastive Learning Loss.

---

> ### Author Response · Authors · 2023-11-23
> **Response to Reviewer CAg3 - W1, W2, W3**
>
> **W1. The dataset lacks a detailed description...**
>
> **A1**. We thank the reviewer for pointing out this problem. The detailed descriptions of datasets, including size, classes, split, preprocessing, and other basic information, are included in Appendix B of the revised manuscript.
>
> **W2. All six methods of experimental comparison rely on adversarially training a sensitive attribute inference model and lack the ability to compare state-of-the-art dp-based methods...**
>
> **A2**. We thank the reviewer for pointing out this related work. However, as stated in Section 2 of our original manuscript, our method aims at providing Information-theoretic Privacy, instead of defending against Membership Inference Attacks as Differential Privacy (DP) does. These are two distinct privacy notions in the literature (Hsu et al., 2021). Thus our method is not comparable with DP-based works. Specifically, Differential Privacy for deep learning generally focuses on creating a randomized optimization mechanism, so that the distributions of learned model parameters are similar for adjacent subsets of the original dataset [Abadi et al. (2016), Zhang et al., (2018), Sun et al., (2020)]. For Information-theoretic Privacy, on the other hand, the goal is to learn a data transformation, so that the information of sensitive attributes contained in the transformed data can be constrained.
>
> References:
>
> - Hsiang Hsu, Natalia Martinez, Martin Bertran, Guillermo Sapiro, and Flavio P Calmon. A survey on statistical, information, and estimation—theoretic views on privacy. IEEE BITS the Information Theory Magazine, 1(1):45–56, 2021
> - Abadi, Martin, et al. "Deep learning with differential privacy." Proceedings of the 2016 ACM SIGSAC conference on computer and communications security. 2016.
> - Zhang, Xinyang, Shouling Ji, and Ting Wang. "Differentially private releasing via deep generative model (technical report)." arXiv preprint arXiv:1801.01594 (2018).
> - Sun, Mingxuan, Qing Wang, and Zicheng Liu. "Human action image generation with differential privacy." 2020 IEEE International Conference on Multimedia and Expo (ICME). IEEE, 2020.
>
> **W3. Lack of comparison with a state-of-the-art method...**
>
> **A3**. We thank the reviewer for pointing out this related work. STPrivacy (Li et al., 2023) proposed an insightful framework for privacy-preserving action recognition (PPAR) for 3D video data, which combined sparsification (removing privacy-leaking action-irrelevant tubelets from ViT input) and anonymization (removing privacy information adversarially by maximizing CE loss). STPrivacy was specially tailored for video data and ViT model, overcoming the dynamic utility loss and privacy leakage of previous frame-based PPAR methods. However, this specialization also hindered its generalizability to other domains, which made it inapplicable to our settings. However, as a remedy, we compared our method with two baseline methods mentioned in the STPrivacy paper, namely PPDAR (Wu et al., 2020) (named VITA in the STPrivacy paper) and SPAct (Dave et al., 2022). The results of PPDAR is shown in Table 1, 2, 4, 5, 7 in our paper. The results of SPAct is shown in the Table 1 below. We can observe that MaSS achieves slightly lower NAG on digit compared to SPAct, but significantly lower NAG on all sensitive attributes (up to 5\%), which shows that our method can achieve a better trade-off between suppression and preservation. However, please note that SPAct did not consider preserving unannotated useful attributes.
>
> **Table 1**: Comparison of the classification accuracy and NAG between MaSS and SPAct on AudioMNIST. We suppress gender, accent, age, id, while preserve digit as annotated useful attribute.
>
> | Method | gender ($\downarrow$) | accent ($\downarrow$) | age ($\downarrow$) | ID ($\downarrow$) | digit ($\uparrow$) |
> |---|---|---|---|---|---|
> |No suppression| 0.9962 (1.0000) | 0.9843 (1.0000) | 0.9657 (1.0000) | 0.9808 (1.0000) | 0.9975 (1.0000) |
> |Guessing| 0.8000 (0.0000) | 0.6833 (0.0000) | 0.1667 (0.0000) | 0.0167 (0.0000) | 0.1000 (0.0000) |
> | SPAct | 0.8087 (0.0442) | 0.6833 (0.0001) | 0.1753 (0.0108) | 0.0707 (0.0560) | 0.9948 (0.9970) |
> | MaSS | 0.8000 (0.0000) | 0.6833 (0.0000) | 0.1662 (0.0000) | 0.0183 (0.0017) | 0.9933 (0.9953) |
>
> References:
>
> - Li, Ming, et al. "STPrivacy: Spatio-Temporal Tubelet Sparsification and Anonymization for Privacy-preserving Action Recognition." arXiv preprint arXiv:2301.03046 (2023).
>
> - Wu, Zhenyu, et al. "Privacy-preserving deep action recognition: An adversarial learning framework and a new dataset." IEEE Transactions on Pattern Analysis and Machine Intelligence 44.4 (2020): 2126-2139.
>
> - Dave, Ishan Rajendrakumar, Chen Chen, and Mubarak Shah. "Spact: Self-supervised privacy preservation for action recognition." Proceedings of the IEEE/CVF Conference on Computer Vision and Pattern Recognition. 2022.

---

> ### Author Response · Authors · 2023-11-23
> **Response to Reviewer CAg3 - Q**
>
> **Q. The authors need provide more detailed explanations and justifications for their proposed techniques...**
>
> **A**. InfoNCE contrastive learning loss (Oord et al., 2018) is a classical contrastive learning loss, which learns useful representations of data by making the representations of positive samples (similar or related samples) closer while pushing the representations of negative samples further apart from the anchor. The sampling strategy in our framework is as follows. Suppose we have ${K+1}$ samples $ \\{ x _i \\}  _{i=1}^{K+1}$ in a mini-batch. We first pass them through the feature extractor $P _\eta(F|X)$ and data transformation module to sample a batch of $ \\{ f _i \\}  _{i=1}^{K+1}$ and $ \\{ x' _i \\}  _{i=1}^{K+1}$ respectively. Then suppose we choose the $j$-th feature $f _j$ as the anchor. Then the corresponding $x' _j$ would be designated as a positive sample, and all other $x' _{i \neq j}$ are  designated as negative samples. After sampling, we calculate the contrastive learning loss as Equation 14 in our paper. For training stability, in our implementation each of the $K+1$ features in a batch are used as an anchor once and then averaged. An analogous strategy was used when anchors are chosen from $x'$.
>
> InfoNCE Contrastive learning is suitable in our design because it is a proper approximation to $I(F;X')$ (and $I(X;X')$), so that it can be used to maximize $I(F;X')$ thus protect unannotated useful attributes in the transformed data. Contrastive learning loss may help with preserving annotated useful attributes as well. But the annotated useful attributes preserving module is taking the primary responsibility, by ensuring the predictability of useful attributes from the transformed data.
>
> Besides, contrastive learning loss also succeeds in that it does not assume the distribution of $X, X'$. It can be universally applied to images, language, sensor signals, etc, with few adjustments. Another important merit of contrastive learning is its advanced empirical performance. We conducted an ablation experiment where we replaced the contrastive learning loss with MSE reconstruction loss (denoted as MaSS-MSE). The results are shown in Table 2 and Table 3. MaSS-NF means MaSS trained without unannotated useful attributes preservation. We can observe that the NAG of digit is significantly lower without unannotated attribute preservation. We can also observe that MaSS achieves higher NAG on digit than MaSS-MSE, as well as lower NAG on other sensitive attributes, which demonstrates the strong empirical performance of contrastive learning loss compared to MSE reconstruction loss.
>
> **Table 2**: Comparison of the classification accuracy and NAG between MaSS and ablations on AudioMNIST. We suppress gender, accent, age, ID, while preserve digit as if an unannotated attribute.
>
> | Method | gender ($\downarrow$) | accent ($\downarrow$) | age ($\downarrow$) | ID ($\downarrow$) | digit ($\uparrow$) |
> |---|---|---|---|---|---|
> | No suppression| 0.9962 (1.0000) | 0.9843 (1.0000) | 0.9657 (1.0000) | 0.9808 (1.0000) | 0.9975 (1.0000) |
> |Guessing| 0.8000 (0.0000) | 0.6833 (0.0000) | 0.1667 (0.0000) | 0.0167 (0.0000) | 0.1000 (0.0000) |
> | MaSS-NF | 0.8000 (0.0000) | 0.6833 (0.0001) | 0.1658 (0.0000) | 0.0152 (0.0000) | 0.2657 (0.1846) |
> | MaSS-MSE | 0.8002 (0.0008) | 0.6833 (0.0001) | 0.1683 (0.0020) | 0.0462 (0.0306) | 0.9542 (0.9517) |
> | MaSS | 0.8000 (0.0000) | 0.6833 (0.0000) | 0.1667 (0.0000) | 0.0195 (0.0029) | **0.9683 (0.9675)** |
>
> **Table 3**: Comparison of the classification accuracy and NAG between MaSS and ablations on MotionSense.  We suppress gender, ID, while preserve activity as if unannotated useful attribute.
>
> | Method | gender ($\downarrow$) | ID ($\downarrow$) | activity ($\uparrow$) |
> |---|---|---|---|
> | No suppression| 0.9026 (1.0000) | 0.9817 (1.0000) | 0.9764 (1.0000) |
> |Guessing| 0.0533 (0.0000) | 0.5699 (0.0000) | 0.4663 (0.0000) |
> | MaSS-NF| 0.0508 (0.0000) | 0.5699 (0.0000) | 0.8374 (0.7275) |
> | MaSS-MSE | 0.0754 (0.0260) | 0.5734 (0.0085) | 0.8823 (0.8156) |
> | MaSS | 0.0555 (0.0026) | 0.5686 (0.0000) | **0.9242 (0.8977)** |
>
>
>
> Reference:
>
> - Aaron van den Oord, Yazhe Li, and Oriol Vinyals. Representation learning with contrastive predictive coding. arXiv preprint arXiv:1807.03748, 2018.

---

### Official Review · Reviewer_YVvf · 2023-10-31

**Soundness:** 3 good
**Presentation:** 3 good
**Contribution:** 3 good
**Rating:** 6
**Confidence:** 3

**Summary:**

The paper presents a novel approach, referred to as MASS (Multi-Attribute Selective Suppression), which addresses the challenge of privacy protection in the context of large-scale datasets used for machine learning. It introduces a formal information-theoretic definition for utility-preserving privacy protection and offers a data-driven, learnable data transformation framework. This framework enables the selective suppression of sensitive attributes while preserving other useful attributes, regardless of whether they are known in advance or explicitly annotated. The paper includes rigorous theoretical analyses of the operational bounds of the proposed framework and conducts extensive experimental evaluations across diverse modalities, such as facial images, voice audio clips, and motion sensor signals. The results demonstrate the effectiveness and generalizability of MASS across different tasks and configurations.

**Strengths:**

1. One of the notable strengths of this paper is the introduction of a formal information-theoretic definition for utility-preserving privacy protection. This theoretical foundation provides a solid framework for addressing privacy concerns in large-scale datasets, contributing to the theoretical underpinning of data privacy solutions.

2. The proposal of a data-driven learnable data transformation framework is innovative. This approach allows for the selective suppression of sensitive attributes, enhancing privacy protection while preserving the utility of the data.

3. The comprehensive experimental evaluations across various data modalities, including facial images, voice audio clips, and motion sensor signals, highlight the generalizability of the MASS framework. This breadth of experimentation underlines its versatility and applicability to a wide range of real-world scenarios.

**Weaknesses:**

1. The paper introduces an interesting concept in Theorem 3.1, highlighting the importance of mutual information constraints 'm' and 'n' in the context of privacy and utility trade-offs. However, it is essential to note that the experiments lack a corresponding exploration of these constraints. These constraints likely play a pivotal role in balancing sensitive attribute accuracy and useful attribute accuracy. The reported experimental results suggest that MaSS might not simultaneously achieve the best performance for both types of attributes. Therefore, it is recommended to conduct experiments with varying constraints to gain a deeper understanding of their impact.

2. The theoretical argument presented in Theorem 3.2 may raise some questions. Specifically, the relevance of unannotated useful attributes to the tasks is highlighted, which could vary across different scenarios. The paper should address this issue and provide an ablation study of the contrastive learning module to support the claims made in Theorem 3.2. This would provide stronger evidence and clarity regarding the relationship between learned attributes and sensitive attributes.

3. Clarification is needed on how the positive and negative samples for the InfoNCE loss are determined. Figure 3 suggests that both positive and negative samples come from the transformed data X', but the paper should explain how these samples are chosen, given that the anchor sample is the original data X.

4. Regarding the evaluation of sensitive attribute accuracy, it appears that the accuracy of the adversarial classifier is used. It is recommended to consider the approach of training a classifier from scratch on the transformed data, similar to the methodology employed for calculating useful attribute accuracy.

5. To provide a more comprehensive assessment of the proposed method, the paper should include comparisons with recent baselines, such as SPAct (CVPR 2022 [1]).

6. The topic of concept removal for generative models, while not a central focus, could be related to this paper's context. It is suggested to discuss concept removal in the related works section to provide a broader perspective on the field and to highlight the paper's contributions in relation to existing research.

[1] Dave, Ishan Rajendrakumar, Chen Chen, and Mubarak Shah. "Spact: Self-supervised privacy preservation for action recognition." Proceedings of the IEEE/CVF Conference on Computer Vision and Pattern Recognition. 2022.

**Questions:**

Please see weaknesses

---

> ### Author Response · Authors · 2023-11-23
> **Response to Reviewer YVvf - Q1, Q2, Q3**
>
> **Q1. The paper introduces an interesting concept in Theorem 3.1, highlighting the importance of mutual information constraints 'm' and 'n' in the context of privacy and utility trade-offs...**
>
> **A1**. We conducted experiments to show the effect of varying constraint, in which we took gender, accent, age and ID as sensitive attributes and digit as an annotated useful attribute on the AudioMNIST dataset. We fixed $m=0$ for gender, accent and age and $n=2.3$ for digit (its maximal value). Then we varied $m$ for ID from $0$ to $1.46$ (its maximal value). The results are shown in Table 1. We can observe that as $m$ increases, the constraint is gradually loosened, which results in the gradually increasing accuracy and NAG for ID.
>
> **Table 1**: Varying the suppression constraint $m$ on AudioMNIST. We suppress gender, accent, age, ID, while preserve digit as if an annotated useful attribute.
>
> | Method | gender ($\downarrow$) | accent ($\downarrow$) | age ($\downarrow$) | ID ($\downarrow$) | digit ($\uparrow$) |
> |---|---|---|---|---|---|
> |No suppression| 0.9962 (1.0000) | 0.9843 (1.0000) | 0.9657 (1.0000) | 0.9808 (1.0000) | 0.9975 (1.0000) |
> |Guessing| 0.8000 (0.0000) | 0.6833 (0.0000) | 0.1667 (0.0000) | 0.0167 (0.0000) | 0.1000 (0.0000) |
> |MaSS $m_{id}$=0.0| 0.8000 (0.0000) | 0.6833 (0.0000) | 0.1662 (0.0000) | 0.0183 (0.0017) | 0.9933 (0.9953) |
> |MaSS $m_{id}$=0.3| 0.8000 (0.0000) | 0.6833 (0.0000) | 0.1665 (0.0000) | 0.0598 (0.0447) | 0.9938 (0.9959) |
> |MaSS $m_{id}$=0.6| 0.8000 (0.0000) | 0.6833 (0.0000) | 0.1668 (0.0002) | 0.1120 (0.0988) | 0.9940 (0.9961) |
> |MaSS $m_{id}$=0.9| 0.8000 (0.0000) | 0.6833 (0.0000) | 0.1670 (0.0004) | 0.1493 (0.1376) | 0.9937 (0.9957) |
> |MaSS $m_{id}$=1.2| 0.8000 (0.0000) | 0.6833 (0.0000) | 0.1667 (0.0000) | 0.1963 (0.1863) | 0.9928 (0.9948) |
> |MaSS $m_{id}$=1.46| 0.8000 (0.0000) | 0.6833 (0.0000) | 0.1667 (0.0000) | 0.2597 (0.2520) | 0.9937 (0.9957) |
>
>
> **Q2. The theoretical argument presented in Theorem 3.2 may raise some questions...**
>
> **A2**. Theorem 3.2 provided an operational upper bound for the learning objective $I(X';F)$, namely $I(X';F) \leq H(X|S _i) + m _i$. The detailed proof can be found in Appendix A.2. The value of the upper bound  ($H(X|S _i) + m _i$) is independent from the parameters of our model, which means that the unannotated useful attributes preservation module can at most preserve $H(X|S _i) + m _i$ nats (Naperian bits) of information of $F$ in $X'$.
>
>
> Besides, $H(X|S _i) + m _i$ is indeed dependent on the dataset and the task. For an extreme example, in a dataset where $X$ is fully decided by $S _i$ ($H(X|S _i)=0$) and $m _i$ is set to 0, then we can have $I(X';F) \leq 0$, which means we can not preserve any information of $F$ in $X'$. On the other hand, if $X$ is conditionally independent from $S _i$ ($H(X|S _i)=H(X)$), we can have $I(X';F) \leq H(X) + m _i$, which is already guaranteed by Data Processing Inequity $I(X';F) \leq I(X'; X) \leq H(X)$. Therefore in this case, Theorem 3.2 no longer casts any additional constraint on $I(X';F)$.
>
> **Q3. Clarification is needed on how the positive and negative samples for the InfoNCE loss are determined...**
>
> **A3**. The sampling strategy in our framework is as follows: Suppose we have ${K+1}$ samples $ \\{ x _i \\}  _{i=1}^{K+1}$ in a mini-batch. We first pass them through the feature extractor $P _\eta(F|X)$ and data transformation module to sample a batch of $ \\{ f _i \\}  _{i=1}^{K+1}$ and $ \\{ x' _i \\}  _{i=1}^{K+1}$ respectively. Then suppose we choose the $j$-th feature $f _j$ as the anchor. Then the corresponding $x' _j$ would be designated as the positive sample, and all other $x' _{i \neq j}$ are  designated as negative samples.  For training stability, in our implementation each of $K+1$ features in a batch are used as an anchor once and then averaged. An analogous strategy was used when anchors are chosen from $x'$.

---

> > ### Author Response · Authors · 2023-11-23
> > **Response to Reviewer YVvf - Q4, Q5, Q6**
> >
> > **Q4. Regarding the evaluation of sensitive attribute accuracy, it appears that the accuracy of the adversarial classifier is used...**
> >
> > **A4**. Although we would not release the labels of sensitive attributes to the public, here for demonstration we make an assumption that the attacker can access the ground truth labels of sensitive attributes as an oracle and retrain the discriminator on the transformed data. The results are shown in Table 2. We can observe that, using MaSS, the accuracy of the retrained discriminator is higher than adversarial discriminator but is still significantly lower than the discriminator trained using the original data. However, since we only intend to release the transformed data without the sensitive attributes labels, we chose to use the accuracy of the adversarially trained discriminator as the quantitative metric.
> >
> > **Table 2**: Comparison of the accuracy and NAG between a trained-from-scratch discriminator and adversarial discriminator on the Adience dataset.
> >
> > | Method | gender($\downarrow$)  |
> > | -- | -- |
> > | No suppression    | 0.9774 (1.0000) |
> > | Guessing | 0.5240 (0.0000) |
> > | MaSS (discriminator retrained with oracle)   | 0.6029 (0.1740)|
> > | MaSS (adversarial discriminator)   | 0.5240 (0.0000) |
> >
> >
> > **Q5. To provide a more comprehensive assessment of the proposed method, the paper should include comparisons with recent baselines, such as SPAct (CVPR 2022 [1]).**
> >
> > **A5**. SPAct did not consider preserving unannotated useful attributes. Therefore for a fair comparison we considered the scenario where we only had annotated attributes. Results are shown in Table 3. We can observe that MaSS achieves slightly lower NAG on digit compared with SPAct, but significantly lower NAG on all sensitive attributes (up to 5\%), which suggests that our method can achieve a better trade-off between suppression and preservation.
> >
> >
> > **Table 3**: Comparison of the classification accuracy and NAG between MaSS and SPAct on AudioMNIST. We suppress gender, accent, age, id, while preserve digit as annotated useful attribute.
> >
> > | Method | gender ($\downarrow$) | accent ($\downarrow$) | age ($\downarrow$) | ID ($\downarrow$) | digit ($\uparrow$) |
> > |---|---|---|---|---|---|
> > |No suppression| 0.9962 (1.0000) | 0.9843 (1.0000) | 0.9657 (1.0000) | 0.9808 (1.0000) | 0.9975 (1.0000) |
> > |Guessing| 0.8000 (0.0000) | 0.6833 (0.0000) | 0.1667 (0.0000) | 0.0167 (0.0000) | 0.1000 (0.0000) |
> > | SPAct | 0.8087 (0.0442) | 0.6833 (0.0001) | 0.1753 (0.0108) | 0.0707 (0.0560) | 0.9948 (0.9970) |
> > | MaSS | 0.8000 (0.0000) | 0.6833 (0.0000) | 0.1662 (0.0000) | 0.0183 (0.0017) | 0.9933 (0.9953) |
> >
> >
> >
> > **Q6. The topic of concept removal for generative models, while not a central focus, could be related to this paper's context...**
> >
> > **A6**. We thank the reviewer for pointing out the connection. The similarity between our method and concept removal for generative models lies in that, both of the two fields are aimed at suppressing an undesired attribute from the transformed/generated data.
> >
> > However, the key difference is that, concept removal focuses on removing a particular undesired class of an attribute (e.g., one may view an experiment in Gandikota et al. (2023) as removing the class of "Van Gogh" from the attribute of "artistic style", while other artistic style classes are allowed to persist.). In comparison, our proposed framework focuses on removing an entire undesired attribute altogether from the transformed data (e.g., no artistic styles can be recovered from the transformed data.).
> >
> > Reference:
> >
> > - Gandikota, Rohit, et al. "Erasing concepts from diffusion models." arXiv preprint arXiv:2303.07345 (2023).

---

### Official Review · Reviewer_AsFx · 2023-11-05

**Soundness:** 3 good
**Presentation:** 3 good
**Contribution:** 3 good
**Rating:** 6
**Confidence:** 3

**Summary:**

This paper proposes an information theorem based multi-attribute selective suppression (MaSS) to solve the problem of highlighting the utility attributes while suppressing the private attributes. The introduced problem is interesting and important, as privacy becomes a central concern for many applications. The paper presents a clear pipeline leveraging three effort stream lines: (1) sensitive attribute suppression, (2) annotated useful attribute preservation and (3) unannotated useful attribute preservation. The optimization are then jointly optimized.

**Strengths:**

1. The paper study into an interesting and practical problem by discussing the limitations and compare to the literature approaches, providing a sufficient background for the problem study.

2. The paper provides a theoretical analysis from the information theory perspective, showing the relationship between utility and sensitive attributes.

3. The paper presents a clear and tractable learning scheme to achieve the three stream lines.

4. There are extensive experimental comparison against the representative literature methods and some state-of-the-arts. Consistently advantageous results demonstrate the method’s effectiveness.

**Weaknesses:**

1. For the sensitive attribute suppression, the objective is to minimize the the expectation of the entropy between P(Si|x) and P_phi(Si|X’). Drawing connection to adversarial learning, it tries to push P_phi(Si|x’) close to P(Si|x) by pushing the discriminator cannot tell the difference between the two.

Firstly, the paper lacks the interpretation of their proposed method, and drawing the connection to the literature method, e.g., adversarial learning. It would be good the authors can conduct an in-depth analysis comparing the literature to the proposal in this paper, and further highlight the method’s novelty.

2. For unannotated useful attribute, depending on the definition of the problem, the setting will be different from the other method. In the paper of experiments, the authors mention “ALR, BDQ and PPDAR overlook the preservation of unannotated useful attributes”.

It could be that those methods, from their problem definition and setting, they do not consider so termed “unannotated useful attributes” into their framework. But one cannot say it is the limitation or fault of those methods. In the most fair way, because of setting difference, this paper should compare to only those considering “unannotated useful attributes”. Please carefully phrase the comparison to other methods.

3. Still, for those methods that are sharing exactly the same setting, e.g., GAP and MSDA, from technical frame design, what is the difference? I noticed there is some slight comparison, e.g., arguing that some of the methods lack theoretical analysis. This is the advantage of this paper. But other than that, if there is an empirical design that is exactly the same as this paper, this paper will only go for the theoretical contribution.

Thus, please provide a towards thorough comparison to those literature under the same setting, which will be helpful to claim the method contribution and novelty.

**Questions:**

Please refer to weakness session for detail.

---

> ### Author Response · Authors · 2023-11-23
> **Response to Reviewer AsFx - Q1, Q2, Q3**
>
> **Q1. For the sensitive attribute suppression, the objective is to minimize the the expectation of the entropy between P(Si|x) and P_phi(Si|X’)...**
>
> **A1**. The relationship between our sensitive attributes suppression and adversarial learning is as follows:
>
> First, our sensitive attributes suppression can be viewed as one type of adversarial learning. When we use linear relaxation and $I(X'; S _i) \geq m _i$, it can be derived from Equation 6 and Equation 8 that our sensitive attributes inference model $\phi _i$ is trained **adversarially** with the data transformation module by solving a min-max optimization as:
>
> $ \max _\theta \min _{\phi _i} \mathbb{E} _{P(X)P _\theta(X'|X)} [H(P(S _i|X),P _{\phi _i}(S _i|X'))], $
>
> where $\phi _i$ is trained to narrow the gap between $P(S _i|X)$ and $P _{\phi _i}(S _i|X')$, while $\theta$ is trained to widen the gap between $P(S _i|X)$ and $P _{\phi _i}(S _i|X')$.
>
> However, the objective of our min-max optimization is different from other adversarial learning methods. In adversarial training for adversarial robustness, the objective is the cross entropy between prediction and groun truth (Bai et al., 2021). In classical GAN (Goodfellow et al., 2014), the objective is the cross entropy loss of differentiating real and fake samples; in WGAN (Arjovski et al., 2017), the objective is the estimated Wasserstein distance between real and fake data distributions.
>
>
> References:
>
> - Bai, Tao, et al. "Recent advances in adversarial training for adversarial robustness." arXiv preprint arXiv:2102.01356 (2021).
> - Goodfellow, Ian, et al. "Generative adversarial nets." Advances in neural information processing systems 27 (2014).
> - Arjovsky, Martin, Soumith Chintala, and Léon Bottou. "Wasserstein generative adversarial networks." International conference on machine learning. PMLR, 2017.
>
>
>
> **Q2. For unannotated useful attribute, depending on the definition of the problem, the setting will be different from the other method...**
>
> **A2**. We thank the reviewer for pointing out this problem. We have revised the discussion in our revised manuscript to establish a fair and clear view on all baselines. For example, we changed the original term "overlook" to "do not consider".
>
> **Q3. Still, for those methods that are sharing exactly the same setting, e.g., GAP and MSDA, from technical frame design, what is the difference?..**
>
> **A3**. We thank the reviewer for the suggestion. The key difference of technical design between our method and GAP, MSDA is that, both GAP and MSDA utilize a heuristic MSE reconstruction loss for preserving unannotated useful attributes, which has inferior empirical performance, and can not be readily generalized to other datasets (e.g. language, etc.). On the other hand, the contrastive learning loss used in our method does not assume the distribution of $X, X'$, and therefore can be universally applied to images, language, sensor signals, etc, with few adjustments to yield good performance.
>
>
> References:
>
> - Huang, Chong, et al. "Generative adversarial privacy." arXiv preprint arXiv:1807.05306 (2018).
>
> - Malekzadeh, Mohammad, et al. "Mobile sensor data anonymization." Proceedings of the international conference on internet of things design and implementation. 2019.

---

### Official Review · Reviewer_wuPW · 2023-11-09

**Soundness:** 2 fair
**Presentation:** 2 fair
**Contribution:** 2 fair
**Rating:** 3
**Confidence:** 5

**Summary:**

The paper proposes a method for learning censored data transformations providing guarantees on the quantity of information about both annotated and unannotated features preserved by said transformations. The authors motivate the method using an information-theoretic calculus and establish operational bounds on the entailed objective. Practically, censoring of the designated sensitive attributes is achieved through a standard (margin-based) adversarial information-minimisation (infomin) procedure; useful annotated and unannotated information is preserved through the use of supervised and contrastive learning, respectively. The authors conduct experiments on datasets covering a range of modalities in AudioMNIST, Motion Sense, and Adience and demonstrate favourable performance of their method relative to the baseline suite.

**Strengths:**

- Figures and tables are well-put-together; Figures 1 and 2 illustrate the problem setup and methodological pipeline, respectively, in an easily digestible manner -- one can understand the essence of the method based on its illustration alone.
- Experiments cover a good range of datasets and configurations.
- Proofs and implications of the consequent theoretical statements are easy to follow.
- The problem under consideration is well-motivated and clearly formulated.
- Reasonable assortment of baseline methods and strong empirical performance of the proposed method relative to these. Experimental setups are described with clarity.
- Good contextualization w.r.t. prior work, with clear delineation of the subtle but differentiating qualities of the current work.

**Weaknesses:**

- The paper is limited in terms of novelty. The main contribution of the paper seems to be in its proposal to preserve of *unannotated* features using a self-supervised-learning objective yet this idea of maximising $\mathcal{I}(X; \tilde{X})$, understanding $\tilde{X}$ to be some representation generally, is certainly not novel in and of itself (vide Madras et al., 2018 in the context of the adjacent field of fair-representation learning); the method chosen used to accomplish this maximisation seems, to me, largely incidental -- one can simply view the contrastive loss as an alternative reconstruction loss.
- The paper proposes learning a data transformation instead of a representation but the codomain of the transformation is another seemingly incidental factor given that interpretability does not appear to be a major concern, based on the narrative and analysis; indeed, in order to compute the contrastive learning objective in a space in which distances are meaningful, the transformed and original samples ultimately have to be embedded in such a representation anyway. The learning of a data transformation instead of a low-dimensional representation leads to a method that is more complicated than seemingly need be, and, moreover, is a design choice that comes at a steep computational cost -- computing all losses in representation space would be much more efficient though there may be some sound theoretical barrier to do doing so.
- The mathematical formalism is confusing and, at times, unrigorous. Random variables and their realisations are seemingly conflated without reference to the abuse: mutual information, $\mathcal{I}(\cdot; \cdot)$ is defined between pairs of random variables, not their empirical counterparts. While there is an argument to be made that such overloading is conventional and remedied by the context, I don't think that the latter is entirely satisifed here, especially with their being no express mention of this overloading being adopted throughout the paper.
- While their meaning, as analogues of $X_p$ and $X_n$, respectively, can be easily inferred, $F_p$ and $F_n$, appearing in Eq.17, seem to be missing explicit definitions. The explanation given in Sec. 4.4,both textually and notationally, is generally muddled considering that the method amounts to SimCLR with the original and transformed samples acting as anchors and positive pairs.
- Why compute cross-entropy terms w.r.t. the estimates of $P(U_i|X)$ and $P(S_i|X)$ as opposed to simply using the (degenerate ground-truth distribution (the annotations) used in the fitting of those estimates? There may be good reason for it but there should clear explanation given for why this choice is unprincipled, should that indeed be the case.
- The quality of the writing, in terms of clarity and structure, could generally do with improvement.
- Lack of ablation studies, such as those investigating the influence of the loss prefactor.
- No discussion of the practical challenges entailed by adversarial infomin (vide Song and Shmatikov, 2021, for instance).


### References
Madras D, Creager E, Pitassi T, Zemel R. Learning adversarially fair and transferable representations. In International Conference on Machine Learning 2018 Jul 3 (pp. 3384-3393). PMLR.

Song C, Shmatikov V. Overlearning Reveals Sensitive Attributes. In8th International Conference on Learning Representations, ICLR 2020 2020 Jan.

**Questions:**

See Weaknesses

---

> ### Author Response · Authors · 2023-11-23
> **Response to Reviewer wuPW - Q1**
>
> **Q1. The paper is limited in terms of novelty...**
>
> **A1**. Our key novelty lies in our proposal of utility preservation, especially for the unannotated attributes from an **information theoretic** perspective, as opposed to purely **heuristic-driven** methods.
>
> Although similar ideas of preserving generic features and maximizing $I(X;X')$ can be found in the literature, these existing approaches have noticeable limitation in that they only preserve generic features by minimizing a heuristic reconstruction loss, which is limited in that, 1) it is oftentimes not general enough to be applied in different datasets as they might require different ways to measure the reconstruction loss, for example, MSE for image and Cross-entropy for language; and 2) its performance is not supported by information theoretic analysis.
>
> For example, Madras et al. (2018) and Edwards et al. (2015) based their fair representation learning methods on an MSE reconstruction loss. Although they provided insightful and sound analyses on the fairness of the learned representation, they did not embed their reconstruction loss into a theoretical framework. In addition, Huang et al. (2019) and Malekzadeh et al. (2019) targeted similar tasks as ours, but resorted to a heuristic MSE reconstruction loss. On the contrary, in our work we proposed to approach the preservation of the generic features from an information theoretic point of view, and accordingly derived a contrastive learning-based framework.
>
> Contrastive learning plays a vital role in our design because InfoNCE contrastive learning loss is not only a proper "reconstruction loss", but also a proper approximation to $I(F;X')$ (and $I(X;X')$), which is consistent with our problem definition. Besides, contrastive learning loss does not assume the distribution of $X, X'$. It can be universally applied to images, language, sensor signals, etc, with few adjustments. Another important merit of contrastive learning is its advanced empirical performance. We conducted an ablation experiment where we replaced the contrastive learning loss with MSE reconstruction loss (denoted as MaSS-MSE). The results are shown in Table 1 and Table 2. MaSS-NF means MaSS trained without unannotated useful attributes preservation. We can observe that the NAG (normalized accuracy gain) of digit is significantly lower without unannotated attribute preservation. We can also observe that MaSS achieved higher NAG on digit than MaSS-MSE, as well as lower NAG on other sensitive attributes, which demonstrate the strong empirical performance of contrastive learning loss compared to MSE reconstruction loss.
>
> **Table 1**: Comparison of the classification accuracy and NAG between MaSS and ablations on AudioMNIST. We suppress gender, accent, age, ID, while preserve digit as if an unannotated attribute.
>
> | Method | gender ($\downarrow$) | accent ($\downarrow$) | age ($\downarrow$) | ID ($\downarrow$) | digit ($\uparrow$) |
> |---|---|---|---|---|---|
> | No suppression| 0.9962 (1.0000) | 0.9843 (1.0000) | 0.9657 (1.0000) | 0.9808 (1.0000) | 0.9975 (1.0000) |
> |Guessing| 0.8000 (0.0000) | 0.6833 (0.0000) | 0.1667 (0.0000) | 0.0167 (0.0000) | 0.1000 (0.0000) |
> | MaSS-NF | 0.8000 (0.0000) | 0.6833 (0.0001) | 0.1658 (0.0000) | 0.0152 (0.0000) | 0.2657 (0.1846) |
> | MaSS-MSE | 0.8002 (0.0008) | 0.6833 (0.0001) | 0.1683 (0.0020) | 0.0462 (0.0306) | 0.9542 (0.9517) |
> | MaSS | 0.8000 (0.0000) | 0.6833 (0.0000) | 0.1667 (0.0000) | 0.0195 (0.0029) | **0.9683 (0.9675)** |
>
> **Table 2**: Comparison of the classification accuracy and NAG between MaSS and ablations on MotionSense.  We suppress gender, ID, while preserve activity as if unannotated useful attribute.
>
> | Method | gender ($\downarrow$) | ID ($\downarrow$) | activity ($\uparrow$) |
> |---|---|---|---|
> | No suppression| 0.9026 (1.0000) | 0.9817 (1.0000) | 0.9764 (1.0000) |
> |Guessing| 0.0533 (0.0000) | 0.5699 (0.0000) | 0.4663 (0.0000) |
> | MaSS-NF| 0.0508 (0.0000) | 0.5699 (0.0000) | 0.8374 (0.7275) |
> | MaSS-MSE | 0.0754 (0.0260) | 0.5734 (0.0085) | 0.8823 (0.8156) |
> | MaSS | 0.0555 (0.0026) | 0.5686 (0.0000) | **0.9242 (0.8977)** |
>
> References:
>
> - Madras, David, et al. "Learning adversarially fair and transferable representations." International Conference on Machine Learning. PMLR, 2018.
> - Edwards, Harrison, and Amos Storkey. "Censoring representations with an adversary." arXiv preprint arXiv:1511.05897 (2015).
> - Huang, Chong, et al. "Generative adversarial privacy." arXiv preprint arXiv:1807.05306 (2018).
> - Malekzadeh, Mohammad, et al. "Mobile sensor data anonymization." Proceedings of the international conference on internet of things design and implementation. 2019.

---

> ### Author Response · Authors · 2023-11-23
> **Response to Reviewer wuPW - Q2, Q3, Q4, Q5**
>
> **Q2. The paper proposes learning a data transformation instead of a representation but the codomain of the transformation is another seemingly incidental factor given that interpretability does not appear to be a major concern...**
>
> **A2**. This paper is targeted at learning a data transformation instead of a compact representation for the following reasons:
>
> First, transformed data can deliver better **reusability for the community**. For example, users can utilize the transformed data for training generative models. For another case, users can develop their own possibly more generalizable feature extractors on the transformed data. Data reusability is crucial for the proliferation of Machine Learning community.
>
> Second, transformed data can be readily used by **pre-trained off-the-shelf models**, whereas compact representations can only be used by models specially trained for them. To empirically support this claim, we ran an experiment of facial landmark detection on both the original and the transformed Adience datasets based on torchlm (https://github.com/DefTruth/torchlm), with the PIPNet (Jin et al., 2021) with ResNet18 backbone. This model generates 98 landmarks for each facial image. We then took the landmarks detected from the original images as the ground truth and evaluated the  landmarks detected from the transformed images, using normalized mean error (NME) (Jin et al., 2021) as the quantitative metric. The results are shown in Table 3. We can observe that the NME between the transformed Adience and the original Adience is comparable to the NME between the original WLFW dataset (Wu et al., 2018) and the ground truth label, which suggests that the transformed dataset can be accurately exploited by a pre-trained model.
>
> **Table 3**: The NME(\%) of PIPNet between transformed Adience and original Adience, in comparison to the NME(\%) of PIPNet between original WLFW dataset and ground truth label. The comparable performance showed that transformed Adience dataset can be accurately exploited by pre-trained PIPNet. On the contrary, compact representations would not be compatible with the pre-trained PIPNet.
>
> | Dataset | WLFW (original vs. label) | Adience (transformed vs. original) |
> |---|---|---|
> | NME $(\downarrow)$  |3.94|3.30|
>
> References:
>
> - Haibo Jin, Shengcai Liao, and Ling Shao. 2021. Pixel-in-Pixel Net: Towards Efficient Facial Landmark Detection in the Wild. Int. J. Comput. Vision 129, 12 (Dec 2021), 3174–3194. https://doi.org/10.1007/s11263-021-01521-4
> - Wayne Wu, Chen Qian, Shuo Yang, Quan Wang, Yici Cai, Qiang Zhou. 2018 CVPR. Look at Boundary: A Boundary-Aware Face Alignment Algorithm
>
>
> **Q3. The mathematical formalism is confusing and, at times, unrigorous...**
>
> **A3**. We thank the reviewer for pointing out these problems. We adjusted the notations in our revised paper. Specifically, we use uppercase (e.g. $X,F$) for random variables and lowercase (e.g. $x,f$) for their realizations. We also changed the original function name $f$ to $\mathcal{F}$ to correct the notation collision.
>
> **Q4. While their meaning, as analogues of $X _p$ and $X _n$, respectively, can be easily inferred...**
>
> **A4**.  We thank the reviewer for pointing out these problems. We included the definitions of $f _p$ and $f _n$ in the revised paper.
>
> **Q5. Why compute cross-entropy terms w.r.t. the estimates of $P(U _j|X)$ and $P(S _i|X)$ as opposed to simply using the (degenerate ground-truth distribution (the annotations) used in the fitting of those estimates?...**
>
> **A5**. We would like to point out that we did already use this degenerate distribution to compute cross entropy in our original manuscript, as stated in Section 3, Section 4.2 and Section 4.3. We updated the description in the revision for improved clarity.

---

> ### Author Response · Authors · 2023-11-23
> **Response to Reviewer wuPW - Q6, Q7, Q8**
>
> **Q6. The quality of the writing, in terms of clarity and structure, could generally do with improvement.**
>
> **A6**. We thank the reviewer for the suggestion. We have updated our manuscript to improve its clarity and structure.
>
> **Q7. Lack of ablation studies, such as those investigating the influence of the loss prefactor.**
>
> **A7**. Apart from the ablation we provided in Table 1 and 2 examining the unannotated attributes preservation module, we also conducted experiments to show the effect of varying the constraint on sensitive attributes suppression ($m$). We took gender, accent, age and ID as sensitive attributes and digit as an annotated useful attribute on the AudioMNIST dataset. We fixed $m=0$ for gender, accent and age and $n=2.3$ for digit (its maximal value). Then we varied $m$ for ID from $0$ to $1.46$ (its maximal value). The results are shown in Table 4. We can observe that as $m$ increases, the constraint is gradually loosened, which results in the gradually increasing accuracy and NAG for ID.
>
> **Table 4**: Varying the suppression constraint $m$ on AudioMNIST. We suppress gender, accent, age, ID, while preserve digit as if an annotated useful attribute.
>
> | Method | gender ($\downarrow$) | accent ($\downarrow$) | age ($\downarrow$) | ID ($\downarrow$) | digit ($\uparrow$) |
> |---|---|---|---|---|---|
> |No suppression| 0.9962 (1.0000) | 0.9843 (1.0000) | 0.9657 (1.0000) | 0.9808 (1.0000) | 0.9975 (1.0000) |
> |Guessing| 0.8000 (0.0000) | 0.6833 (0.0000) | 0.1667 (0.0000) | 0.0167 (0.0000) | 0.1000 (0.0000) |
> |MaSS $m_{id}$=0.0| 0.8000 (0.0000) | 0.6833 (0.0000) | 0.1662 (0.0000) | 0.0183 (0.0017) | 0.9933 (0.9953) |
> |MaSS $m_{id}$=0.3| 0.8000 (0.0000) | 0.6833 (0.0000) | 0.1665 (0.0000) | 0.0598 (0.0447) | 0.9938 (0.9959) |
> |MaSS $m_{id}$=0.6| 0.8000 (0.0000) | 0.6833 (0.0000) | 0.1668 (0.0002) | 0.1120 (0.0988) | 0.9940 (0.9961) |
> |MaSS $m_{id}$=0.9| 0.8000 (0.0000) | 0.6833 (0.0000) | 0.1670 (0.0004) | 0.1493 (0.1376) | 0.9937 (0.9957) |
> |MaSS $m_{id}$=1.2| 0.8000 (0.0000) | 0.6833 (0.0000) | 0.1667 (0.0000) | 0.1963 (0.1863) | 0.9928 (0.9948) |
> |MaSS $m_{id}$=1.46| 0.8000 (0.0000) | 0.6833 (0.0000) | 0.1667 (0.0000) | 0.2597 (0.2520) | 0.9937 (0.9957) |
>
>
> **Q8. No discussion of the practical challenges entailed by adversarial infomin (vide Song and Shmatikov, 2021, for instance).**
>
> **A8**. We thank the reviewer for pointing out this related work. For both de-censoring and re-purposing mentioned in Song et al. (2020), they require the access to the trained black box feature extractor (or, equivalently, the data transformation module in our case). However, the objective of our work is to release the transformed data in order to unleash its utility without the risk of leaking its sensitive attributes, whereas the transformation module itself will not be released.
>
> Reference:
>
> - Song, Congzheng, and Vitaly Shmatikov. "Overlearning reveals sensitive attributes." arXiv preprint arXiv:1905.11742 (2019).

---

### Author Response · Authors · 2023-11-23
**General Response to Reviewers**

We thank the reviewers for their questions and suggestions. We appreciate their recognition of our strengths, such as

- The problem under consideration is well-motivated and clearly formulated (Reviewer wuPW);
- The proposal of a data-driven learnable data transformation framework is innovative (Reviewer YVvf);
- The theoretical foundation provides a solid framework for addressing privacy concerns in large-scale datasets (Reviewer YVvf);
- There are extensive experimental comparison against representative literature methods (Reviewer AsFx); and
- The clarity of the paper is commendable, and the paper is well structured (Reviewer CAg3).

In our responses, we addressed all reviewers' concerns with detailed explanations and experiments. In particular, we conducted 5 new experiments:

- Ablation study of unannotated useful attributes preservation module (Table 1, 2 in response to Reviewer wuPW and Table 2, 3 in response to Reviewer CAg3);
- Usability of transformed data by pre-trained model (Table 3 in response to Reviewer wuPW);
- Ablation study of varying constraint for the sensitive attribute (Table 4 in response to Reviewer wuPW and Table 1 in response to Reviewer YVvf);
- Accuracy of a discriminator for sensitive attribute re-trained with oracle (Table 2 in response to Reviewer YVvf); and
- Comparison with SPAct (Table 3 in response to Reviewer YVvf and Table 1 in response to Reviewer CAg3).

---

### Meta-Review · Area_Chair_fFhQ · 2023-12-23

**Metareview:**

The paper proposes a formal information-theoretic definition for utility-preserving privacy protection. It introduces a data-driven learnable data transformation framework that is capable of selectively suppressing sensitive attributes from target datasets while preserving the other useful attributes, regardless of whether or not they are known in advance or explicitly annotated for preservation.

All reviewers identified critical concerns with the novelty and significance of the contributions (theoretical contributions, missing baseline comparisons), and experimental results. Among others, the authors ignore a large set of prior art representation learning for fairness, including Madras et al. These are not just related papers, they were designed for the sensitive attribute suppression, and should be carefully compared against. There is also a concern that more revisions are needed at this time to make the paper palatable; and new experiments and insights produced during rebuttal would need another round of reviews to evaluate the overall effectiveness and the claims in this work.

After extended deliberation in the end the decision was made to reject the paper. Ultimately, the weaknesses in the evaluation and the somewhat incremental contribution were just too hard to overlook. We hope the reviews are useful for improving and revising the paper.

**Justification For Why Not Higher Score:**

More revisions are needed at this time to make the paper palatable; and new experiments and insights produced during rebuttal would need another round of reviews to evaluate the overall effectiveness and the claims in this work.

**Justification For Why Not Lower Score:**

N/A

---

### Decision · Program_Chairs · 2024-01-16

Reject